# Transcriptome analysis illuminates the nature of the intracellular interaction in a vertebrate-algal symbiosis

**John A Burns[1,2]\*, Huanjia Zhang[3], Elizabeth Hill[3], Eunsoo Kim[1,2], Ryan Kerney[3]\***

[1]Division of Invertebrate Zoology, American Museum of Natural History, New York, United States; [2]Sackler Institute for Comparative Genomics, American Museum of Natural History, New York, United States; [3]Department of Biology, Gettysburg College, Gettysburg, United States

**Abstract** During embryonic development, cells of the green alga *Oophila amblystomatis* enter cells of the salamander *Ambystoma maculatum* forming an endosymbiosis. Here, using *de novo* dual-RNA seq, we compared the host salamander cells that harbored intracellular algae to those without algae and the algae inside the animal cells to those in the egg capsule. This two-by-two-way analysis revealed that intracellular algae exhibit hallmarks of cellular stress and undergo a striking metabolic shift from oxidative metabolism to fermentation. Culturing experiments with the alga showed that *host* glutamine may be utilized by the algal *endosymbiont* as a primary nitrogen source. Transcriptional changes in salamander cells suggest an innate immune response to the alga, with potential attenuation of NF-κB, and metabolic alterations indicative of modulation of insulin sensitivity. In stark contrast to its algal endosymbiont, the salamander cells did not exhibit major stress responses, suggesting that the host cell experience is neutral or beneficial.

**\*For correspondence:** jburns@ amnh.org (JAB); rkerney@ gettysburg.edu (RK)

**Competing interests:** The authors declare that no competing interests exist.

## Introduction

All vertebrates have a 'microbiome' that includes mutualist ecto-symbionts living in close association with, but not within, their cells (*Douglas, 2010*). The most substantial vertebrate ecto-symbioses occur in the colon and small intestine and are implicated in physiological processes such as nutrient absorption from undigested complex carbohydrates (*Ley et al., 2008*; *Krajmalnik-Brown et al., 2012*). Known endosymbioses in vertebrates, where microbial cells live within the vertebrate cells, are almost exclusively parasitic, causing diseases such as malaria, toxoplasmosis, and chytridomycosis (*Douglas, 2010*; *Sibley, 2004*; *Davidson et al., 2003*). Currently, there is only a single exception. The green alga *Oophila amblystomatis* enters the cells of the salamander *Ambystoma maculatum* during early development (*Kerney et al., 2011*), and co-culture experiments show that the algae consistently benefit the salamander embryo hosts (*Small et al., 2014*; *Graham et al., 2013*; *Pinder and Friet, 1994*).

There is a long history of experimentation on the *ecto*symbiotic association between *O. amblystomatis* and *A. maculatum*: where the alga populates salamander egg capsules that contain developing embryos (*Small et al., 2014*; *Gilbert, 1944*). In the ectosymbiosis, the alga appears to benefit from nitrogenous waste excreted by the developing embryo while providing periodic oxygen and photosynthate to the microenvironment of the embryo's egg capsule, aiding salamander development (*Small et al., 2014*; *Graham et al., 2013*; *Gilbert, 1944*). However, the intracellular association between these two organisms was only recently recognized (*Kerney et al., 2011*), 122 years after the first published description of green salamander egg masses (*Orr, 1888*).

**eLife digest** Throughout the natural world, when different species form a close association, it is known as a symbiosis. One species can depend on another for food, defense against predators or even for reproduction. Corals, for example, incorporate single-celled algae into their own cells. The algae photosynthesize, harnessing energy from sunlight to make sugars and other molecules that also feed the coral cells. In return, corals protect the algae from the environment and provide them with the materials they need for photosynthesis. This type of relationship where one organism lives inside another species is called an endosymbiosis.

In animals with a backbone, endosymbioses with a photosynthetic organism are rare. There is only one known example so far, which is between a green alga called *Oophila amblystomatis* and the spotted salamander, *Ambystoma maculatum.* The female spotted salamander deposits her eggs in pools of water, and algae enter the eggs, proliferate, and later invade tissues and cells of the developing embryos. However, it is not understood how similar the interaction between the alga and the salamander is to that in coral-algal symbioses, or whether it is rather more similar to a parasitic infection.

Burns et al. now address this question by comparing salamander cells harboring algae to those that lacked algae. A technique called RNA-Seq was used to characterize the changes in gene activity that take place in both organisms during the endosymbiosis. The results show that algae inside salamander cells are stressed and they change the way in which they make energy. Instead of carrying out photosynthesis to produce food for the salamander host – as happens in coral-algal interactions – *Oophila amblystomatis* is fighting to adapt to its new environment and switches to a less efficient energy producing pathway known as fermentation.

Burns et al. found that, in striking contrast to the alga, affected salamander cells do not show signs of stress. Instead several genes that are known to suppress immune responses against foreign invaders are expressed to high levels. This may explain how salamander cells are able to tolerate algae inside them.

The next challenge is to understand how the alga enters salamander cells. The current work identified some potential routes of entry, and follow up studies are now needed to explore those possibilities.

Perhaps the most intensively studied endosymbiosis between photosynthetic microbes and non-photosynthetic animal hosts is the facultative mutualistic interactions between various invertebrate cnidarians (i.e. corals and sea anemones) and dinoflagellate endosymbionts (*Davy et al., 2012*). Such interactions provide the host animals the ability to obtain energy through photosynthesis. Cnidarian-dinoflagellate endosymbioses involve a number of physiological changes in both the host and photo-symbiont on a cellular level. The host animal tends to exhibit a tempered immune response to the ingressing cells (modulated by the host, symbiont, or both) (*Detournay et al., 2012*), and to express genes necessary for transferring nutrients to the symbiont (*Lehnert et al., 2014*), receiving nutrients from the symbiont (*Lehnert et al., 2014*), and use of the endosymbiont-derived metabolites (*Lehnert et al., 2014*). Less is known about adaptations of the endosymbiotic cells, but they can include modified osmoregulation (*Mayfield and Gates, 2007*), export of nutrients to the host cell (*Lin et al., 2015*), and physical changes such as loss of flagella (*Muller-Parker et al., 2015*).

In this study, we used a dual RNA-Seq approach on wild-collected *A. maculatum* salamander embryos and their endosymbiont alga *O. amblystomatis* to characterize the transcriptomic changes that occur in both organisms during this unique endosymbiosis. We isolated free-swimming algal cells living within the egg capsule ('intracapsular environment', triplicate sampling), salamander cells that did not contain algae (N = 50 cells per replicate, quadruplicate sampling), and salamander cells containing intracellular algae (N = 50 cells per replicate, quadruplicate sampling) from the same individuals. We identified differentially expressed genes in both organisms attributed to the intracellular association. The algal endosymbiont undergoes drastic changes in metabolism, displaying signs of cellular stress, fermentation, and decreased nutrient transport, while the host salamander cell

displays a limited innate immune response and changes to nutrient sensing, but does not appear to invoke cell stress responses such as apoptosis or autophagy.

## Results

### Cell isolation, mRNA sequencing, and de-novo assembly

Ectosymbiotic, intra-capsular algal cells were isolated from egg capsules with a syringe (*Figure 1a*). Individual *A. maculatum* cells were manually separated into groups of 50 cells with or without intracellular algal symbionts (*Figure 1a,b*). Total RNA was extracted from *A. maculatum* cells or from intra-capsular algal samples, and converted to cDNA (*Figure 1c*). A test for contaminating mRNA from *A. maculatum* lysed during dissociation was shown to be negative (*Figure 1—figure supplement 1*) A total evidence assembly contained all reads from all samples (n = 3 intra-capsular algal samples from three different eggs; salamander cells with and without algae from n = 4 individual salamander embryos). This was followed by homology and abundance filtering (*Figure 1—figure supplements 2*, *3* and *4*), producing 46,549 *A. maculatum* and 6,726 *O. amblystomatis* genes that were used in differential expression analysis.

The salamander and algal transcriptomes were tested for completeness using BUSCO (Benchmarking Universal Single-Copy Orthologs) analysis (*Simão et al., 2015*). The final filtered algal assembly contained 31% (130/429) of eukaryote BUSCOs, reduced, due to limitations of sequencing depth in intracellular algal samples, from 47% (199/429) for algal genes in the total evidence assembly. For comparison, a de-novo transcriptome assembly from *O. amblystomatis* cultured in replete media, contained 79% (336/429) of eukaryote BUSCOs. This is comparable to the *Chlamydomonas reinhardtii* transcriptome, containing 74% (316/429) of eukaryote BUSCOs. The algal transcriptome generated from the wild collected samples, however, was prepared using a different library preparation protocol (SMARTer cDNA synthesis followed by Nextera-XT library preparation). This was chosen for the low cell numbers of salamander cells with and without endosymbionts. The transcriptome derived from the cultured alga was sequenced from a TrueSeq library preparation. This was chosen due to relatively large quantities of RNA from lab cultured algal strains. The algal transcriptome from the wild collected total evidence assembly (SMARTer cDNA synthesis and Nextera-XT library preparation) was found to be missing as much as about 40% of the total algal transcriptome, likely due to GC-content biases introduced during library preparation (*Figure 1—figure supplement 5a and b*) (*Lan et al., 2015*). The incompleteness of the transcriptome did not affect inference of differentially expressed genes from the set of 6,726 found in all algal samples. However, the low-cell count library preparation protocol did limit the sensitivity of our algal analysis in that we could not draw inferences from genes that were not present in the wild-collected algal libraries.

The final filtered *A. maculatum* transcriptome assembly contained 88% (375/429) of eukaryote BUSCOs and 69% (2,078/3,023) of vertebrate BUSCOs. For comparison, the *A. mexicanum* transcriptome assembly (*Smith et al., 2005*; *Voss et al., 2015*; *Baddar et al., 2015*; *Voss, 2016*) contained 89% (381/429) of eukaryote BUSCOs and 65% (1,953/3,023) of vertebrate BUSCOs. The SMARTer cDNA synthesis followed by Nextera-XT library prep did not exclude expected salamander transcripts. This is likely due to the low GC content of these RNAs, with a median 43% GC content compared to the algal transcript's median GC content of 62% (*Figure 1—figure supplement 5c*).

### Differential expression (DE) analysis

Among the 6,726 *O. amblystomatis* genes available for DE analysis, 277 were significantly differentially expressed with a false discovery rate (FDR) adjusted p-value (*Benjamini and Hochberg, 1995*) of less than 0.05 (*Figure 1d*) between intracellular and intracapsular algae. There were 111 genes with higher expression in intracellular algae and 166 genes with lower expression in intracellular algae. Of those, 56 (50%) of the over expressed genes and 91 (55%) of the under expressed genes were assigned putative functions based on homology to known proteins. The genes were grouped into eighteen broad functional categories (*Table 1*) revealing the response of the alga to the intracellular environment. Intracellular algae exhibit a stress response with over-expression of three heat shock proteins and other indicators of oxidative and osmotic stress, and large metabolic changes compared to freely swimming intracapsular algal cells. The complete list of annotated, differentially expressed alga genes can be found in the file *Table 1—source data 1.*

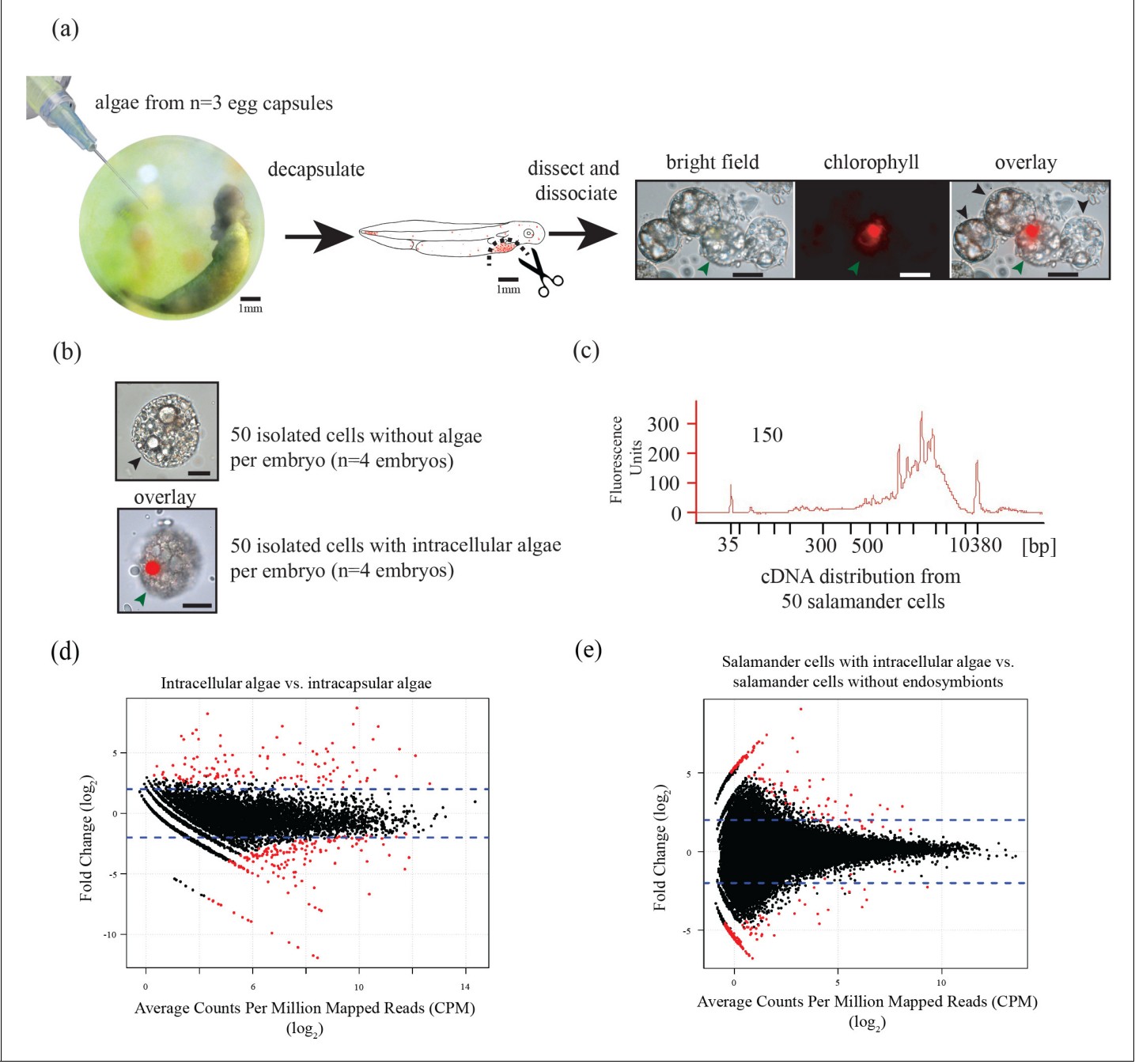

**Figure 1.** Three populations of cells from *A. maculatum* egg capsules containing stage 39 embryos were collected and prepared for mRNA extraction, cDNA sequencing, and differential expression analysis revealing several hundred significantly differentially expressed genes detected for the salamander and alga. (a) Intracapsular algae (Population 1) were removed from intact eggs using a syringe and hypodermic needle (photo credit: Roger Hangarter). Embryos were decapsulated and washed, and the liver diverticulum region (dashed line), containing high concentrations of algae (red dots), was isolated and dissociated into a single cell suspension (illustration adapted from *Harrison, 1969*). The dissociated cells were screened for *A. maculatum* endoderm cells without alga (black arrowheads) and endoderm cells with intracellular alga (green arrowhead). Scale bars on microscope images are 20 µm. (b) Isolated endoderm cell, and isolated endoderm cell with intracellular alga. Scale bars on microscope images are 20 µm. (c) Representative cDNA distribution (bioanalyzer trace) from a population of 50 manually isolated *A. maculatum* endoderm cells. Peaks at 35 bp and 10380 bp are markers. Due to evidence of lysed *A. maculatum* cells observed in the cell suspension fluid after dissociation of *A. maculatum* embryos (debris seen in dissociated *A. maculatum* microscope images in (a) and (b)), that fluid was tested for the presence of contaminating mRNA. mRNA was not detected in the surrounding fluid, *Figure 1—figure supplement 1*. Lower limit abundance thresholds (*Figure 1—figure supplement 2*), and correction for low sequencing depth in intracelluar algal samples (*Figure 1—figure supplement 3*) were implemented to obtain the final gene sets used for differential expression analysis. Depth of sequencing was not biased for *A. maculatum* cell with and without alga samples (*Figure 1—figure*

*Figure 1 continued on next page*

*Figure 1 continued*

*supplement 4*). Library preparation GC bias affected the completeness of the algal transcriptome obtained from intracapsular and intracellular *O. amblystomatis* (*Figure 1—figure supplement 5*). (d and e) Dotplots of log$_2$ fold change vs. expression level. The blue horizontal lines are plus and minus 4-fold change in expression between samples. The red dots are genes with FDR adjusted p-values<0.05, indicating a significant difference in expression level between conditions. (d) Differentially expressed algal transcripts. (e) Differentially expressed salamander transcripts.

The following source data and figure supplements are available for figure 1:

**Source data 1.** Raw counts matrix with counts for all reads mapped to the total evidence assembly (the assembly of all salamander and algal reads from wild-collected samples).

**Source data 2.** List of 6,726 algal gene IDs used in differential expression analysis.

**Source data 3.** List of 46,549 salamander gene IDs used in differential expression analysis.

**Figure supplement 1.** *A. maculatum* cell lysis during embryo dissociation did not contaminate the cell suspension fluid with significant quantities of mRNA.

**Figure supplement 2.** Determining lower limit FPKM thresholds for inclusion in differential expression analysis.

**Figure supplement 3.** Determining a threshold for absence calls in intracellular algal data.

**Figure supplement 4.** Determining threshold for absence calls in salamander data.

**Figure supplement 5.** High GC content algal genes were not detected by the combination of SMARTer cDNA synthesis and Nextera-XT library preparation.

In *A. maculatum*, 46,549 genes were analyzed for differential expression. A total of 300 genes were identified as differentially expressed with an FDR adjusted p-value less than 0.05 (*Figure 1e*). There were 134 genes with higher expression in salamander cells containing intracellular algae and 166 genes with lower expression in those cells. Of those, 74 (55%) of over expressed genes and 71 (43%) of the under expressed genes were assigned putative functional annotations. The genes were grouped into twelve broad functional categories (*Table 2*) reflecting the response of *A. maculatum* cells to the intracellular algae. Transposable elements comprise the largest category of annotated differentially expressed genes (18% of over- and 27% of the under-expressed). Other functional responses include an immune response to the intracellular alga, modulation of the host cell's nutrient sensing, and differential expression of genes related to cell survival and interactions with other cells, including cell-cell adhesion and motility. The complete list of annotated, differentially expressed algal genes can be found in the file *Table 2—source data 1*.

## Phosphate and nitrogen transporters in *O. amblystomatis* are regulated by phosphate and glutamine levels, respectively

Cultures of the symbiotic alga in AF6 media, allowed in vitro testing of algal inorganic phosphate and nitrogen transporter regulation, in response to availability of relevant nutrient sources.

The high affinity phosphate transporter *PHT1-2*, was regulated by extracellular inorganic phosphate concentration in cultured *O. amblystomatis* (*Figure 2a*). The average qPCR expression difference between high (100 µM and above) and low (10 µM and below) phosphate concentrations was 32-fold (p=4.4 $\times$ 10$^{-15}$), which agrees with both the RNA-seq data (25 fold lower expression in the endosymbiotic alga), and estimates of phosphate concentrations in vernal pool water (low micromolar) (*Brodman et al., 2003*; *Carrino-Kyker and Swanson, 2007*) compared to inside amphibian cells (low millimolar) (*Horowitz et al., 1979*; *Burt et al., 1976*). A second phosphate transporter, a chloroplast localized sodium dependent phosphate transport protein 1 (*ANTR1*), was not regulated by extracellular phosphate levels (*Figure 2b*). Its low expression level in the endosymbiotic alga is therefore not likely to be related directly to an increased phosphate level of the host cytoplasm.

Expression of two inorganic nitrogen transporters (ammonium transporter 1-member 2, *AMT1-2* and high-affinity nitrate transporter 2.4, *NRT2.4*) and a urea-proton symporter, *DUR3* was repressed

**Table 1.** Functional classification of the green alga *O. amblystomatis* genes that are differentially expressed during intracellular association with the salamander host.

| Functional Category | # genes | #up | #down |
|---|---|---|---|
| No Homology | 90 | 43 | 47 |
| Conserved Gene with Unknown Function | 37 | 11 | 26 |
| Stress Response | 32 | 14 | 18 |
| Fermentation | 17 | 13 | 4 |
| Electron Transport-Mitochondrial | 6 | 0 | 6 |
| Photosynthesis | 13 | 7 | 6 |
| Ribosomal Proteins | 11 | 1 | 10 |
| Nitrogen Transport | 5 | 0 | 5 |
| Phosphate Transport | 2 | 0 | 2 |
| Other Transport | 12 | 6 | 6 |
| Sulfur Metabolism | 5 | 5 | 0 |
| Lipid Metabolism | 7 | 5 | 2 |
| Other Metabolism | 9 | 0 | 9 |
| Flagellar Apparatus | 4 | 1 | 3 |
| Signaling | 5 | 1 | 4 |
| Transposable Element | 4 | 1 | 3 |
| Glycosylation | 2 | 0 | 2 |
| Other | 13 | 2 | 11 |
| Totals | 277 | 111 | 166 |

**Source data 1.** Differentially expressed algal transcripts, annotations, functional groupings, and expression statistics.

by L-glutamine (*Figure 2c*). Adding 2 mM glutamine, the concentration observed in the cytoplasm of amphibian cells (*Vastag et al., 2011*; *Westermann et al., 2016*, *2012*), to algal cultures induced down-regulation of *AMT1-2* (17-fold, p=3.2 $\times$ $10^{-4}$), *NRT2.4* (7-fold, p=0.013), *DUR3* (278-fold, p=9.5 $\times$ $10^{-10}$). All of these in vitro changes closely match the in vivo expression differences revealed by RNA-seq for the intracellular alga (*Figure 2c*).

## Discussion

### Dual-RNA-Seq of a vertebrate endosymbiont

This study provides the first transcriptomes for *A. maculatum* and *O. amblystomatis* and an in-depth look at gene expression changes of both organisms in their unique endosymbiotic state. The dual-RNA-seq approach has previously been used to investigate intracellular parasitism in vertebrates (*Westermann et al., 2016*, *2012*; *Tierney et al., 2012*). However, our analysis represents the first investigation of a *vertebrate endosymbiosis* where the generalized interaction between the two organisms has consistently been characterized as a mutualism (*Small et al., 2014*; *Gilbert, 1944*; *Bachmann et al., 1986*). Our results also extend dual-RNA-seq methodology to low cell number samples from wild collected, non-model organisms. The transcriptional responses to this cellular association reveals how a vertebrate host responds to an intracellular mutualist and offers insights into the physiological condition of both partners in their endosymbiotic state.

In the host salamander, we identified only a small fraction of the analyzed genes (300/46,549; 0.64%) that are differentially expressed between endosymbiont-bearing vs endosymbiont-free salamander cells. This tempered host response is reminiscent of that of the hosts in coral-dinoflagellate

**Table 2.** Functional classification of the salamander, *A. maculatum,* genes that are differentially expressed when associated with intracellular alga.

| Functional Category | # genes | #up | #down |
| --- | --- | --- | --- |
| No Homology | 155 | 60 | 95 |
| Transposable Element | 69 | 24 | 45 |
| Immune Response | 12 | 11 | 1 |
| Nutrient Sensing | 14 | 7 | 7 |
| Metabolism | 8 | 6 | 2 |
| Adhesion/ECM | 7 | 4 | 3 |
| Proliferation/Survival/ Apoptosis | 7 | 7 | 0 |
| Motility | 5 | 3 | 2 |
| Transcriptional Regulation | 6 | 2 | 4 |
| Cell-Type Specific | 3 | 3 | 0 |
| DNA Repair | 3 | 3 | 0 |
| Others | 11 | 4 | 7 |
| Totals | 300 | 134 | 166 |

Source data 1. Differentially expressed salamander transcripts, annotations, functional groupings, and expression statistics.

endosymbioses; less than 3% of the analyzed genes were shown to be differentially expressed when the host coral was inoculated with and without a symbiosis competent strain of dinoflagellate (*Voolstra et al., 2009*; *Mohamed et al., 2016*). By comparison, the algal response to endosymbiosis from ectosymbiosis was observed to be more pronounced; 4.12% (277/6,726) of the algal genes were differentially expressed, proportionally 6.4 times more genes than in the host salamander. This level of change, nevertheless, is much more subtle when compared to the changes observed between the endosymbiont algal transcriptome and the cultured free-living alga grown in nutrient replete conditions where 40% (2,687/6,726) of the algal genes were differentially expressed.

## Expression differences in intracellular algae

The over-expression of heat shock proteins, autophagy related proteins, and other stress inducible genes reveal hallmarks of stress in the intracellular algae (*Supplementary file 1*). These are undergoing multiple metabolic changes compared to their free-swimming intracapsular counterparts. Intracellular algae parallel the response of the closely related green alga *Chlamydomonas reinhardtii* to low sulfur levels (*Supplementary file 2*) under hypoxia, including gene expression changes consistent with a switch from oxidative to fermentative metabolism (*Supplementary file 3*) (*Nguyen et al., 2008*; *Grossman et al., 2011*). This response, relative to intracapsular algae, includes under-expression of photosystem II core components (*Supplementary file 4*) in the chloroplast and complex I of the electron transport chain in the mitochondrion (*Piruat and López-Barneo, 2005*) (*Supplementary file 5*), along with over-expression of fermentative metabolic pathways that would shuttle pyruvate toward acetyl-CoA, organic acids and alcohols [crucially, over-expression of pyruvate-ferredoxin oxidoreductase (*PFOR*), phosphate acetyltransferase (*PAT*), and aldehyde-alcohol dehydrogenase (*ADHE*)], and potentially produce $H_2$ gas [over-expression of an iron hydrogenase (*HYDA1*)] (*Supplementary file 3*) (*Volgusheva et al., 2013*; *Yang et al., 2013*; *Catalanotti et al., 2013*).

To verify the observed expression differences between intracapsular and intracellular algae, we performed a comparison of expression in the intracellular algae to *O. amblystomatis* gene expression in unialgal culture in nutrient replete media. A complete analysis of differentially expressed genes between *O. amblystomatis* cultured in nutrient replete media and intracellular algae revealed

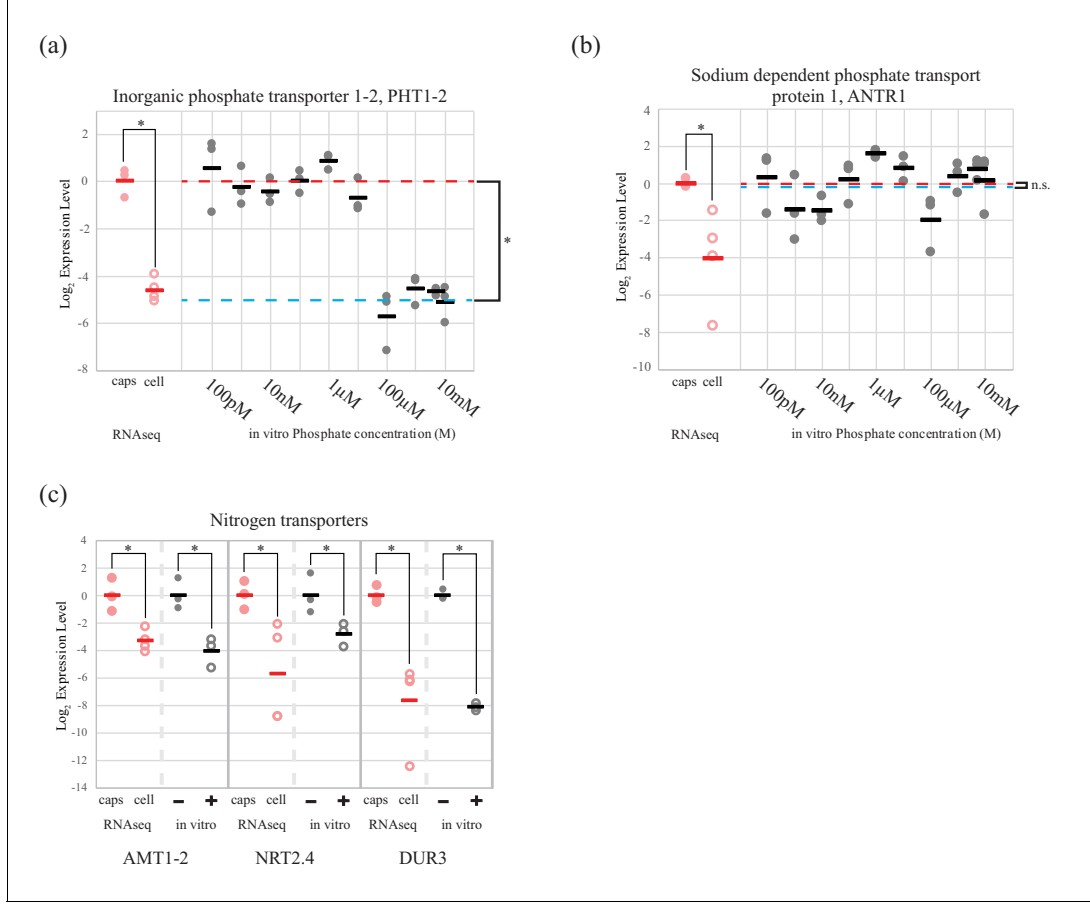

**Figure 2.** An algal phosphate transporter is modulated by inorganic phosphate levels, while nitrogen source transporters are modulated by an organic nitrogen source, glutamine. Normalized measurements from RNAseq data are provided for direct visual comparison of effect sizes in intracellular algae compared to in vitro experiments. Intracapsular alga measurements are 'caps' (filled red circles); intracellular alga measurements are 'cell' (empty red circles). (a) Expression of high affinity phosphate transporter *PhT1-2* mRNA across a range of phosphate concentrations. (b) Expression of chloroplast sodium dependent phosphate transporter *ANTR1* mRNA across a range of phosphate concentrations. In (a) and (b) The red dashed line indicates the average expression of the phosphate transporter in the low phosphate range (100 pM to 1 µM); the blue dashed line indicates the average expression in the high phosphate range (10 µM to 10 mM). (c) Expression of three algal nitrogen transporters in the absence (-) and presence (+) of 2 mM L-glutamine. Data is plotted on a $\log_2$ scale on the y axis, where more negative values indicate lower expression levels. Circles are individual replicates; bars are the average for each experiment. *$p < 0.05$; n.s. indicates no significant difference; the statistical test performed was an ANOVA with contrasts.

The following source data is available for figure 2:

**Source data 1.** Normalized expression levels of algal phosphate transporters.

**Source data 2.** Normalized expression levels of algal nitrogen transporters.

1,805 over-expressed transcripts and 882 under-expressed transcripts in the intracellular algae (indicating 40% of transcripts are differentially expressed, *Figure 3*). A summary of GO terms enriched among the 1,805 genes over-expressed in intracellular algae relative to algae cultured in nutrient replete media confirms an enrichment in fermentation and stress response processes (*Figure 3—figure supplement 1*). Processes enriched among the 882 under-expressed genes are also consistent with low oxygen and stress to the intracellular algae relative to algae cultured in nutrient replete media (*Figure 3—figure supplement 2*).

Specific consideration of the 36 genes demarking the fermentation response in intracellular algae compared to intracapsular algae shows that 21 are similarly significantly differentially expressed when compared to their expression in cultured *O. amblystomatis* from nutrient replete medium (*Figure 4*). Each of these 21 genes are similarly over- or under-expressed in the intracellular-intracapsular

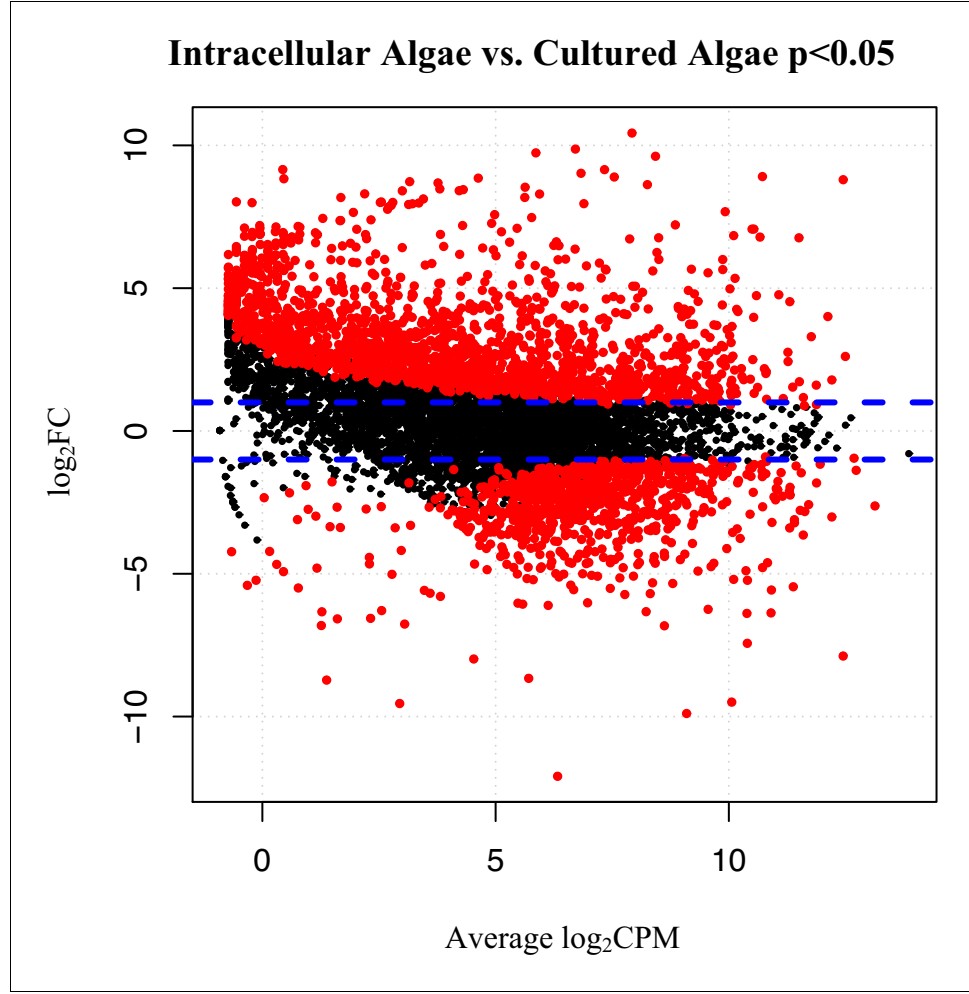

**Figure 3.** Differentially expressed genes between intracellular algae and cultured algae. Red dots indicate significantly differentially expressed genes (FDR < 0.05). Blue dashed lines represent a plus and minus 2-fold difference in expression. There are 1,805 over-expressed genes in intracellular algae and 802 under-expressed genes in intracellular algae in this comparison.

The following source data and figure supplements are available for figure 3:

**Source data 1.** GC content and length of algal genes.
**Figure supplement 1.** REViGO anlysis of GO terms associated with 1805 over-expressed genes in intracellular algae compared to cultured algae.
**Figure supplement 2.** REViGO anlysis of 882 under-expressed genes in intracellular algae compared to cultured algae.

and intracellular-cultured algae comparisons. This includes under-expression of 3 components of the mitochondrial electron transport chain, and consistent over-expression of *PFOR*, *PAT*, *ADHE* and *HYDA1*. Equivalent photosystem II core components are not significantly under-expressed in intracellular algae compared to cultured algae, suggesting that intracapsular algae over-express photosystem II core components, rather than intracellular algae under-expressing them. This may be due to hyperoxic conditions in the intracapsular environment (*Pinder and Friet, 1994*), which could lead to oxidative damage to and rapid turnover of the photosystem II core (*Richter et al., 1990*).

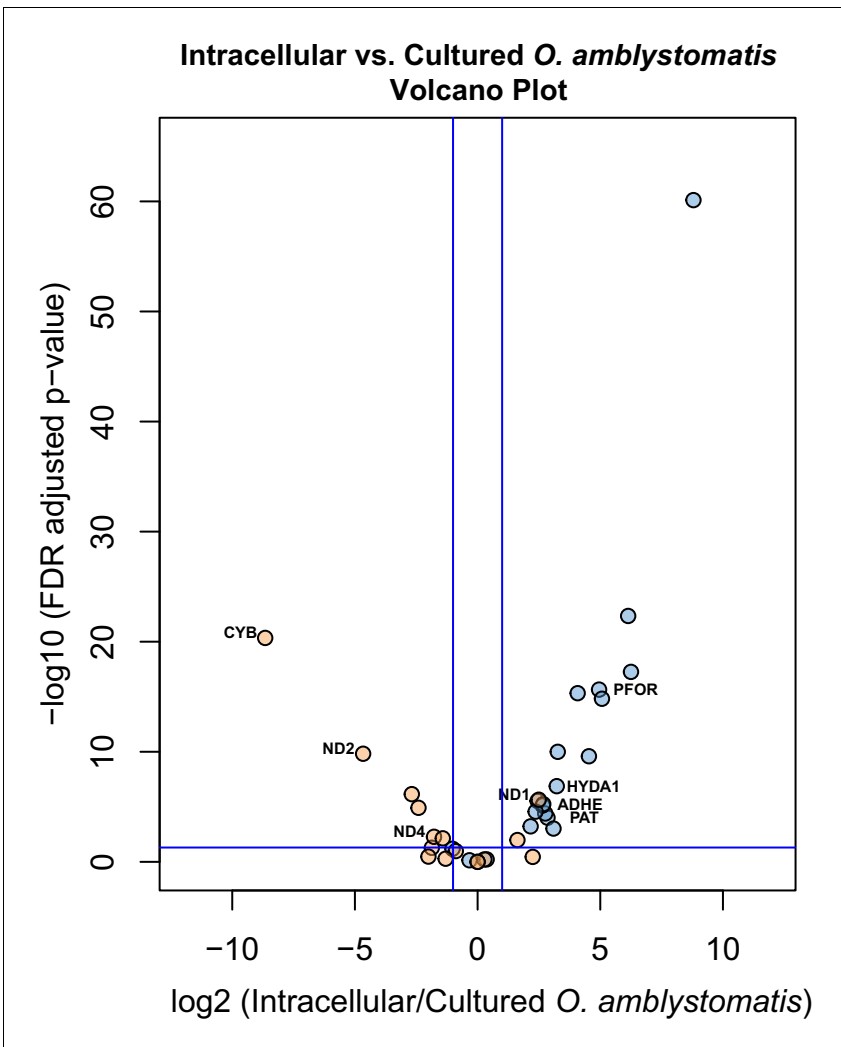

**Figure 4.** Differential expression of fermentation genes in intracellular algae compared to cultured algae. Blue dots are genes that were over-expressed in intracellular algae compared to intracapsular algae. Orange dots are genes that were under-expressed in intracellular algae compared to intracapsular algae. Vertical blue lines represent plus and minus two-fold fold change. The horizontal blue line represents FDR adjusted p-value equal to 0.05. Genes above the horizontal blue line are significantly differentially expressed; genes below the blue line are not. Key fermentation genes, PFOR, HYDA1, ADHE, and PAT are significantly over-expressed in intracellular algae compared to cultured algae, in the same manner as they are over-expressed in intracellular algae compared to intracapsular algae. Several components of complex I of the electron transport chain in the mitochondrion are also significantly under-expressed (CYB, ND2, ND4), though ND1 is over-expressed in intracellular algae compared to cultured algae.

Overall, these metabolic changes suggest that photosynthesis by the alga does not keep up with respirational demand during the endosymbiotic state. In algal fermentation, some photosynthesis components are still active, notably photosystem II (even if it is downregulated) (*Volgusheva et al., 2013*). However, the complement of over- and under-expressed genes in the alga suggests that neither the diffuse oxygen in these tissues nor the oxygen generated by photosynthesis is sufficient to meet the metabolic demands of the algal cell through oxidative phosphorylation. Instead these cells have switched to fermentation. This is potentially attributable in part to the opaque tissues of the embryonic host, which restrict the necessary photons from reaching algal chloroplasts that would allow more oxygen to be generated by the splitting of water.

The occurrence of intracellular fermentation is also supported by decreased starch granules in the intracellular algae and transcriptional changes to algal sulfate transport and sulfur metabolism associated genes. Previous analysis of algal ultrastructure revealed intracellular algae had a significant reduction of starch reserves compared to intracapsular algae (*Kerney et al., 2011*). Intracellular algae had approximately 56% of the starch reserves of intracapsular algae by cross sectional area, which corresponds to around 42% of the starch reserves by volume (*Kerney et al., 2011*). Starch consumption during algal fermentation is well characterized in other Chlamydomonad algae (*Zhang et al., 2002*). Two recent studies observed a reduction of intracellular starch reserves to approximately 50% of their peak levels during long term sulfur deprivation (*Zhang et al., 2002*; *Chader et al., 2009*). The overexpression of a sulfate transporter and taurine catabolic enzymes (*González-Ballester et al., 2010*), along with other transcripts associated with sulfur metabolism (*Figure 3*; *Supplementary file 2*), indicates that fermentation in intracellular *Oophila* coincides with sulfur deprivation and closely matches the consumption of starch found in other fermenting algae.

One discrete class of algal changes entails modifications to nitrogen and phosphorous transporters (*Supplementary file 6*), which are likely attributed to relatively high concentrations of intracellular nutrients compared to the egg-capsule microenvironment (*Brodman et al., 2003*; *Carrino-Kyker and Swanson, 2007*; *Horowitz et al., 1979*; *Burt et al., 1976*; *Vastag et al., 2011*; *Goff and Stein, 1978*). Transcriptional responses, closely matching those seen in our DE analysis, were initiated by mimicking intracellular concentrations of phosphate or glutamine in cultured algal stocks. The algal down regulation of inorganic nitrogen transporters in response to glutamine suggests that the algal endosymbiont is using host glutamine as a nitrogen source. This scenario is supported by the use of glutamine as a sole nitrogen source in other related green algal taxa (*Neilson and Larsson, 1980*). The changes in transporter expression indicate that metabolite concentration differences in an algal cell's microenvironment can account for potentially all of the observed transcriptional differences in the DE analysis. This not only validates our dual-RNA-seq experimental design but suggests mechanisms of niche-dependent transcriptional regulation consistent with other green algae (*Fan et al., 2016*), and the potential acquisition of host-derived glutamine for intracellular algal metabolism.

## Expression differences in salamander cells with algal endosymbionts

There are interesting parallels to both parasitic infections and other known facultative endosymbioses in the salamander transcripts expressed. These include innate immune responses, nutrient sensing, cell motility and apoptosis/survival. The changes in transcript expression reveal a unique cytosolic relationship between these salamander cells and their algal endosymbionts.

## Transposable elements

There are a remarkable number of transposable elements that are differentially expressed between salamander cells with and without algal endosymbionts (*Supplementary file 7*). Their differential expression may be controlled by the transcriptional regulation of nearby co-regulated genes (*Batut et al., 2013*). We posit that the observed differential regulation of transposable elements in this study is a function of *A. maculatum*'s extraordinarily large genome (at around 31 Gb, which is approximately ten times the size of the human genome) (*Licht and Lowcock, 1991*). This large size is likely attributable to a large number of mobile elements in the salamander genome (*Keinath et al., 2015*), which may share regulatory regions with protein coding genes (*Batut et al., 2013*). With a few exceptions, the genes annotated as transposable elements have few detectable RNA transcripts (median counts per million of 1.54 and 0.88 for genes with increased or decreased expression in salamander plus endosymbiont samples, respectively) compared to other, non-transposon, differentially expressed genes (median counts per million of 10.34 and 14.82, respectively). Of the 69 genes with sequence homology to known transposable elements, 32 (46%) have homologs in *A. mexicanum* transcriptomes (*Voss, 2016Voss, 2016*; *Stewart et al., 2013*; *Wu et al., 2013*). The transposable elements are largely (68%) long interspersed nuclear elements (*LINE* retrotransposons), which are typically associated with genome expansions in eukaryotes (*Kidwell, 2002*). Other differentially expressed transposable elements are *PLE* retrotransposons (6%), *LTR* retrotransposons (11%), *DIRS* retrotransposons (6%), and *DNA* transposons (3%).

## Apoptosis and cell survival

Intracellular invasion by a foreign microbe can lead to apoptosis in animal cells (e.g. salmonella [*Monack et al., 1996*; *Kim et al., 1998*] or malaria [*Kakani et al., 2016*]). However, the salamander cells with algal endosymbionts did not show any clear transcriptional signs of apoptosis. The one gene whose primary functional role may be in apoptosis is the salamander transcript (Bcl2-like protein 14, *BCL2L14*) with higher expression in salamander cells with algal endosymbionts, which contains *BCL-2* homology (BH) domains BH3 and BH2 (*Figure 5*). Nonetheless, BCL2L14 has been explicitly shown to *not* be involved in pro-apoptotic regulation (*Giam et al., 2012*). Other genes with higher expression in salamander cells with algal endosymbionts (e.g. olfactomedin-4, *OLFM4*; TNFAIP3-interacting protein 1, *TNIP1*; serine/threonine protein kinase 1, *SGK1*) have demonstrated anti-apoptotoic functions in other animals (*Liu and Rodgers, 2016*; *Ramirez et al., 2012*; *Lang et al., 2010*). In the lab, intra-tissue, and potentially intracellular, algal cells are detected for prolonged periods during development and post hatching in salamander larvae up to Stage-46, and algal DNA was detected in adult tissues (*Kerney et al., 2011*). Eventually, the number of detectable algal cells within the larvae decreases (*Kerney et al., 2011*). This may coincide with the development of the salamander's adaptive immune system, or it could be that the alga stops producing chlorophyll, but are nonetheless maintained within the embryo. There are seven transcripts (e.g. *SGK1*; GDNF receptor alpha-4, *GFRA4*; thymosin beta 4, *TMSB4*) (*Supplementary file 8*) with higher expression in salamander cells with algal symbionts that are linked to cell survival in different physiological contexts including cancer cell survival and proliferation, and neuronal survival during development (*Lang et al., 2010*; *Enokido et al., 1998*; *Bock-Marquette et al., 2004*). Genes from this category may contribute to building a novel network of gene regulation used to maintain these endosymbionts.

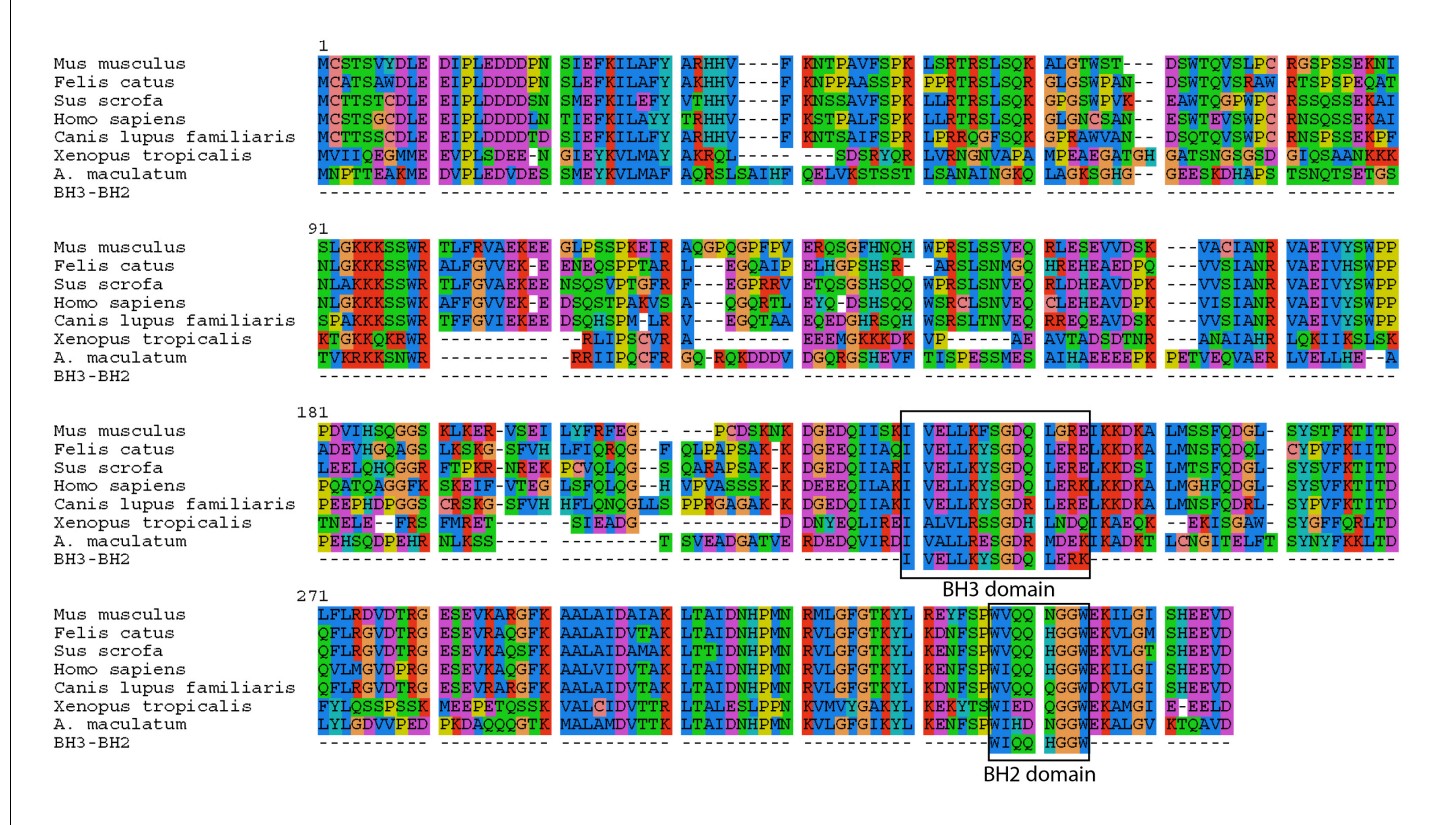

**Figure 5.** *A. maculatum* BCL2L14 protein has both a BH3 and BH2 domain. A multiple alignment of the *A. maculatum* BCL2L14 protein sequence with other organisms reveals a conserved BH3 and BH2 domains (boxed).

## The salamander immune response

Our data reveals a limited immune response of embryonic cells with algal endosymbionts (*Supplementary file 9*). The salamander immune response can largely be categorized as an innate immune response, but it also includes components of an adaptive response (e.g., interleukin 7, *IL7*) that precedes the developmental formation of an adaptive immune system (*Charlemagne and Tournefier, 1998*). Amphibian immunology has, for the most part, been considered a physiological process of larvae and adults (*Savage et al., 2014*). There are no prior transcriptional datasets on embryonic immune responses, although there is a growing interest in amphibian embryo-microbial interactions (*Gomez-Mestre et al., 2006*). Therefore, our study fills a gap pending the availability of comparative data from embryonic pathogens such as the oomycete *Saprolegnia* (*Petrisko et al., 2008*).

The increased expression of pro-inflammatory interleukins and chemokines (e.g. interleukins, *IL-8*; *IL-7*; *IL-1β*; and C-X-C motif chemokine 10, *CXCL10*) parallels the transcriptional response of adult frog skin to the chytrid fungus *Batrachochytridium dendrobatidis* (BD) infection and entry (*Ellison et al., 2014*). Naïve BD infected frogs mount a much more dramatic immune response than these isolated salamander cells (*Savage et al., 2014*; *Ellison et al., 2014*; *Rosenblum et al., 2009*). However, this may, in part, be due to the tissue-level resolution of these studies on chytrid infection, as opposed to the cellular-level resolution of our study, or the comparison of embryos to adults.

Multiple transcriptional differences indicate an increased generation of reactive oxygen species (ROS) in salamander cells containing algae. The NADPH-oxidase family member dual oxidase 1 (*DUOX1*)is involved in the zebrafish intestinal epithelial cell immune response to *Salmonella* infection (*Flores et al., 2010*) and is also more highly expressed in salamander cells with endosymbiotic algae. While this protein is implicated in several physiological processes, its ability to catalyze the generation of reactive oxygen species (ROS) reveals a potentially conserved immune response role in these non-phagocytic embryonic cells. Moreover, we observed under-expression of subunit 6b of cytochrome c oxidase (COX6B1). Reduction of cytochrome c oxidase activity is associated with the generation of ROS through signaling to endoplasmic reticulum NADPH oxidases in yeast (*Leadsham et al., 2013*). Additionally, the downregulation of STEAP4 leads to lower levels of ROS in mouse osteoclasts (*Zhou et al., 2013*), whereas we see upregulation of STEAP4 here. Importantly, the generation of ROS does not rely on the presence of high levels of oxygen in vertebrate cells as ROS can be generated in hypoxic as well as normoxic conditions through a variety of mechanisms (*Kim et al., 2012*; *Nathan and Cunningham-Bussel, 2013*) although there is no indication that the salamander cells are suffering from hypoxia here.

Four genes with high expression during the intracellular symbiosis (*OLFM4*; *TMSB4*; *TNIP1*; and mucin1, *MUC1*) are associated with negative regulation of the NF-κB response in other systems (*Ueno et al., 2008*; *Verstrepen et al., 2009*; *Liu et al., 2010*; *Sosne et al., 2007*). NF-κB is a transcription factor complex that is expressed in all vertebrate cell types, and is involved in a variety of immune responses (*Tato and Hunter, 2002*; *Takada et al., 2005*). Of the possible genes transcriptionally regulated by NF-κB (*Ali and Mann, 2004*; *Gilmore Lab, 2016*), only effectors in the cytokine/chemokine group were observed as over-expressed in salamander cells containing intracellular algae. Our salamander transcriptiomes contain 295 genes that are homologous to downstream targets of NF-κB identified in other systems. Only five are over-expressed here, *IL8*, *IL1b*, *CXCL10* and *TNIP1* (with *TNIP1* also being a negative regulator of NF-κB) (*Supplementary file 10*-sheet 1, 'NFkB_Expr'). A fifth NF-κB response gene, which we have annotated as a trypsin, but which also has strong homology to Granzyme B (associated with apoptosis) was under-expressed in salamander cells hosting endosymbionts (*Supplementary file 10*-sheet 1, 'NFkB_Expr'). Increased expression of genes associated with attenuating NF-κB signal transduction has precedent in other intracellular infections (*Tato and Hunter, 2002*) and possibly symbiotic associations (*McFall-Ngai, 2014*). The over-expressed *OLFM4* and *MUC1* are also implicated in establishing the vaginal microbiome, potentially through their roles in modulating NF-κB signaling (*Doerflinger et al., 2014*; *Fields et al., 2014*). How the algal cell entry may be affecting these genes and whether NF-κB by-pass is involved in algal cell entry or maintenance remains to be determined.

We did not find modified expression of toll-like receptors (TLR's), which detect pathogen-associated molecular patterns. These function in activating an innate immune response to both bacterial and protistan pathogens (*Ashour, 2015*) as well as establishment of gut commensals (*Round et al.,*

*2011*). While TLR's do not require differential regulation for their normal function, established schistosomiasis, entamoeba, trypanosome, and filarial nematode infections all result in *down*-regulation of TLR transcripts (*Ashour, 2015*), and a resulting NF-κB mediated response. In our data, we found 10 salamander genes with homology to various TLRs. None were differentially expressed between salamander cells without and salamander cells with endosymbionts. Further, we examined expression of 151 additional genes associated with the TLR response, and found only one over-expressed gene downstream of TLRs, TNIP1, which is a negative regulator of NF-κB, as discussed above (*Supplementary file 10*-sheet 2, 'TLRs_Expr'). While the lack of differential transcriptional regulation of *TLR's* or their regulators does not preclude their involvement in algal entry response, it is notable in comparison to other parasitic infections where TLR expression is often down regulated (*Ashour, 2015*).

One potential benefit of having an intracellular alga may be to prime the embryo's immune system, without over-activating it, granting the invaded embryos additional protection against the microbial environment outside of the egg capsule. One mechanism for this immune priming may be revealed by relatively increased hepcidin expression in salamander cells with intracellular algae. Elevated hepcidin levels are protective against multiple infections of malaria parasites in mammalian models (*Portugal et al., 2011*) and were shown to enhance resistance to bacterial infection when transgenically over-expressed in zebrafish (*Hsieh et al., 2010*).

## Metabolism and nutrient sensing in *A. maculatum*

In established endosymbioses between invertebrates and algae, the transfer of organic molecules synthesized by the algal partner allows otherwise non-photosynthetic animals to become partial or complete autotrophs. The exchange of photosynthate from symbiont to host is mediated by a range of molecules including sugars, sugar alcohols, and lipids (*Burriesci et al., 2012*; *Kellogg et al., 1983*; *Colombo-Pallotta et al., 2010*). In the endosymbiosis between *A. maculatum* embryos and *O. amblystomatis*, the alga does not appear to be using a canonical photosynthetic process of carbon fixation, oxygen evolution, and sugar production, but is rather metabolizing by fermentation. This metabolic state does not, however, preclude the possibility of metabolite transfer from the intracellular alga to the salamander cells. Under fermentation, the alga may still generate ATP from light energy (*Godaux et al., 2015*), fix carbon (*Godaux et al., 2015*) or use alternate molecules as a carbon source (*Gibbs et al., 1986*). Fermenting algae are also capable of using anabolic reactions to produce sugars and lipids (*Gibbs et al., 1986*). Indeed, the related alga *Chlamydomonas moewusii* excretes glycerol, acetate, and ethanol under anoxic conditions (*Klein and Betz, 1978*). Release of fermentation byproducts such as formate, acetate, and glycerol, or of energy storage molecules such as sugars or lipids into the host cytoplasm could trigger differential expression of nutrient sensing mechanisms within *A. maculatum* cells.

In *A. maculatum* cells with an intracellular alga, differentially expressed genes involved in nutrient sensing (e.g. *STEAP4*; neurosecretory protein VGF, *VGF*; resistin, *RETN*; pyruvate dehydrogenase phosphatase 1, *PDP1*; and calcium/calmodulin-dependent protein kinase 1, *CAM-KK 1*) (*Supplementary file 11*) (*Wellen et al., 2007*; *Petrocchi-Passeri et al., 2015*; *Steppan et al., 2001*; *Jeoung and Harris, 2010*; *Witczak et al., 2007*) are suggestive of altered metabolic flux through catabolic and anabolic pathways, particularly with respect to insulin production and sensitivity. One differentially expressed algal gene that may be implicated in nutrient exchange with salamander cells is an algal gene with homology to Niemann-Pick type C (NPC) proteins, which is more highly expressed in intracellular algae. These proteins are involved in intracellular cholesterol transport, and are potential mediators of lipid transfer in cnidarian-dinoflagellate endosymbioses (*Dani et al., 2014*). Intriguingly, increased expression of this potential sterol sensing gene is observed in our intracellular algal transcripts, whereas in cnidarian-dinoflagellate interactions it is the host that utilizes NPC proteins (*Dani et al., 2014*).

Further metabolic changes include lower expression of maltase-glucoamylase (*MGAM*), an acid phosphatase, and trypsin-like proteins in salamander cells with intracellular algae (*Supplementary file 11*). These changes indicate a reduction in glycogen metabolism (*Barbieri et al., 1967*), and a reduction in the degradation and utilization of yolk platelets that has been shown to be mediated in part by acid phosphatases (*Lemanski and Aldoroty, 1977*). Collectively, these metabolic changes may be induced by detection of metabolites transferred by the alga,

or alternatively, expression changes of these genes might be modulated by autocrine signaling as there is overlap between nutrient sensing and inflammatory responses (*Schenk et al., 2008*).

## Potential mechanisms for algal entry into salamander cells

Lipoprotein-related protein 2 (*LRP2*), which is expressed 5.5 fold higher in invaded salamander cells, is part of a family of multi-ligand receptors that trigger endocytosis (*Supplementary file 11*) (*Fisher and Howie, 2006*). Dual binding of malaria sporozoites to a human LRP receptor and heparin sulfate proteoglycans mediates malaria sporozoite invasion into liver cells (*Shakibaei and Frevert, 1996*). All salamander samples also displayed expression of heparin sulfate basement protein (mean CPM 9.82), though it was not differentially expressed in invaded cells. The alga may have surface proteins that interact with the salamander LRP2 receptor however it does not have a recognizable homolog of the malaria circumsporozoite (CS) protein, which is implicated in interactions with LRP and heparin sulfate proteoglycans (*Shakibaei and Frevert, 1996*). The lack of a recognizable symbiosomal membrane around an intracellular alga (*Kerney et al., 2011*) suggests that if they do indeed enter through endocytosis, they must escape from or degrade the host-derived vesicle. Interestingly, there are two algal lipase/esterase genes and one gene with homology to the bacterial virulence factor streptolysin S that are over-expressed in endosymbiotic algae (55, 11, and 82-fold). Lipase/esterases are known virulence factors in bacterial parasites (*Singh et al., 2010*). These algal lipase/esterases may have a role in algal entry or endosome escape. While the LRP connection to malaria entry is an interesting parallel, the over-expression of *LRP2* may be attributable to other processes in the host cell. For instance LRP2 may be involved in nutrient sensing as it has been implicated in retinol binding protein (RBP) import (Salamander *RBP2* was also more highly expressed in cells with endosymbiotic algae) and vitamin homeostasis (*Christensen et al., 1999*).

The observed increased expression of salamander villin 1 (*VIL1*) in cells with intracellular algae (*Supplementary file 12*) may also reveal a pre-disposition of infected cells for algal entry. Similar predispositions exist in hepatocytes infected with a malaria sporozoite. These express high levels of EphA2 prior to parasite entry, which eventually allows a by-pass of host apoptosis (*Kaushansky et al., 2015*). Villin one is an actin modifying protein that has recently been shown to be required for membrane ruffling and closure following *Salmonella typhimurium* invasion of intestinal epithelial cells (*Lhocine et al., 2015*). This host protein is required for successful pathogen entry and is regulated by the bacterial SptP protein through phosphorylation. Similar membrane ruffling has been observed in regions of algal contact with host alimentary canal epithelium coincident with algal entry (*Kerney et al., 2011*).

## The nature of the endosymbiosis

Whether the alga benefits from this endosymbiotic interaction remains unclear. Similar questions of net 'mutualism' persist for the symbiosis between the bobtail squid *Eprymna scolopes* and the bacterium *Aliivibrio fischeri* (*McFall-Ngai, 2014*) although in both systems microbial cells exhibit specific taxic responses to a developing host, suggesting an adaptive origin of these behaviors. In a previous study, we found evidence consistent with vertical transmission. Algal 18S rDNA was amplified from adult oviducts and Wolffian ducts, and encysted algal cells were found inside the egg capsules of freshly laid eggs using transmission electron microscopy (TEM) (*Kerney et al., 2011*). However, to date, we have not found conclusive evidence for vertical transmission of the alga from one generation to the next. As such, any benefit to the alga in this endosymbiotic interaction remains unknown.

We may speculate that intracellular algae are providing some benefit to its host, as many past light/dark rearing experiments have shown a net benefit to the salamander embryo from their algal symbionts, which presumably included endosymbionts as well (*Pinder and Friet, 1994*; *Gilbert, 1944*; *Bachmann et al., 1986*; *Gilbert, 1944*). Two recent studies have suggested the transfer of photosynthate from intracapsular *Oophila* to the salamander host (*Graham et al., 2013*, *2014*). However oxygenic photosynthesis and fixed carbon photosynthate transfer does not appear to be a significant contribution from intracellular algae to their hosts (*Graham et al., 2013*). Instead these algae appear to be utilizing fermentation, a common response of chalamydomonad algae to sulfur deprivation and hypoxic conditions (*Yang et al., 2015*).

In an intriguing parallel to the metabolic state of intracellular *Oophila*, some obligate intracellular parasites, such as the apicomplexan *Plasmodium* sp., are 'microaerophiles:' these generate most of

their energy through the incomplete oxidation of glucose to lactate, a fermentative process (*Olszewski and Llinás, 2011*). *Plasmodium falciparum* exhibits increased infection and growth at low partial pressures of oxygen (*Ng et al., 2014*; *Scheibel et al., 1979*). Moreover, the observed cell stress response of the alga is reminiscent of that experienced by intracellular apicomplexan parasites (*Bosch et al., 2015*). Mutualist ectosymbionts like the bacteria *Aliivibrio fischeri* also use anaerobic metabolism, including fermentation, and express genes consistent with a response to oxidative stress during their association with the bobtail squid *Eprymna scolopes* (*Thompson et al., 2017*; *Wier et al., 2010*).

The primary fermentation products of chlamydomonad algae include glycerol, ethanol, formate, and acetate, along with smaller amounts of $CO_2$ and $H_2$ (*Catalanotti et al., 2013*). Glycerol is recognized as a major mediator of energy transfer between dinoflagellate photosymbionts and cnidarian hosts (*Davy et al., 2012*). In the context of the salamander-alga endosymbiosis, the related alga *C. reinhardtii* was shown to export glycerol after osmotic shock (*Léon and Galván, 1995*), a condition that intracellular *Oophila* likely experience upon invasion of salamander cells. Additionally, *C. moewusii* excretes glycerol during fermentation (*Klein and Betz, 1978*). Formate and acetate fermentation by-products in bacterial ectosymbionts are known sources of energy and bases for biosynthesis of more complex molecules in animal intestinal cells (*den Besten et al., 2013*; *Karasov and Douglas, 2013*). Moderate ethanol concentrations, below 2 mM, appear to be well tolerated by animal cells (*Castilla et al., 2004*), while higher concentrations become cytotoxic, inducing apoptosis and necrosis (*Castilla et al., 2004*). It is unknown whether intracellular algae are indeed releasing significant quantities of ethanol into host cells, however prior TEM observations (*Kerney et al., 2011*), and our results suggest that ethanol is below cytotoxic levels as we do not see indication of a necrotic or apoptotic host response.

Although the main comparison in this manuscript was between intracapsular algae and intracellular algae, we also considered differentially expressed genes between algae cultured in nutrient replete medium and intracellular algae. The latter comparison supported fermentation in the intracellular algae, but did not indicate over-expression of additional biosynthetic capabilities including enhanced vitamin biosynthesis or the production and export of other metabolites that might be beneficial to the salamander. Whether the intracellular algae are on the positive end of a net host benefit remains uncertain, however it is clear that the algae have an unconventional 'photosymbiont' role.

## Conclusion

To the best of our knowledge there are only two models where the acquisition of horizontally acquired endosymbionts has mechanistic resolution: dinoflagellates in corals and rhizobial bacteria in root nodules. Starting from wild collected samples in a non-model system, we compiled novel transcriptomes of two organisms and revealed gene expression changes associated with their intracellular symbiosis from low cell number samples. These data reveal that life in a vertebrate's cytoplasm induces a stress response in the symbiotic alga. While the alga appears to benefit from high concentrations of phosphate and organic nitrogen sources, our data suggests that the alga is limited in oxygen and sulfur, and is osmotically stressed (*Figure 6*). The salamander appears to recognize the alga as foreign, but does not mount an immune response comparable to what is seen in amphibian-pathogen interactions, and the salamander may be actively repressing important immune regulators such as NF-κB as well as receiving a nutritive benefit from the endosymbiotic alga (*Figure 6*).

Components of the salamander expression profile are relevant to vertebrate interactions with commensal symbionts. For example, three of the genes with higher expression in cells with endosymbionts in this association, *OLFM4, MUC1*, and *IL8* are also up-regulated in human irritable bowel disease (*Gersemann et al., 2012*). Other roles of OLFM4 and MUC1 include negative regulation of NF-κB signal transduction, and interactions with commensal ectosymbionts in humans (*Liu et al., 2010*; *Doerflinger et al., 2014*; *Fields et al., 2014*). The notable absence of other transcripts (e.g. *TLR's*) are indicative of endosymbiont tolerance in this system, in sharp contrast to an expected response to vertebrate pathogens.

As in other endosymbiotic associations, *A. maculatum* and *O. amblystomatis* engage in a unique dialog involving host tolerance of the symbiont and metabolic cross-talk between partners. Distinctive facets of this metabolic cross-talk include fermentation in the endosymbiont as well as phosphate and glutamine acquisition from the host cytoplasm. This study has dramatically expanded our

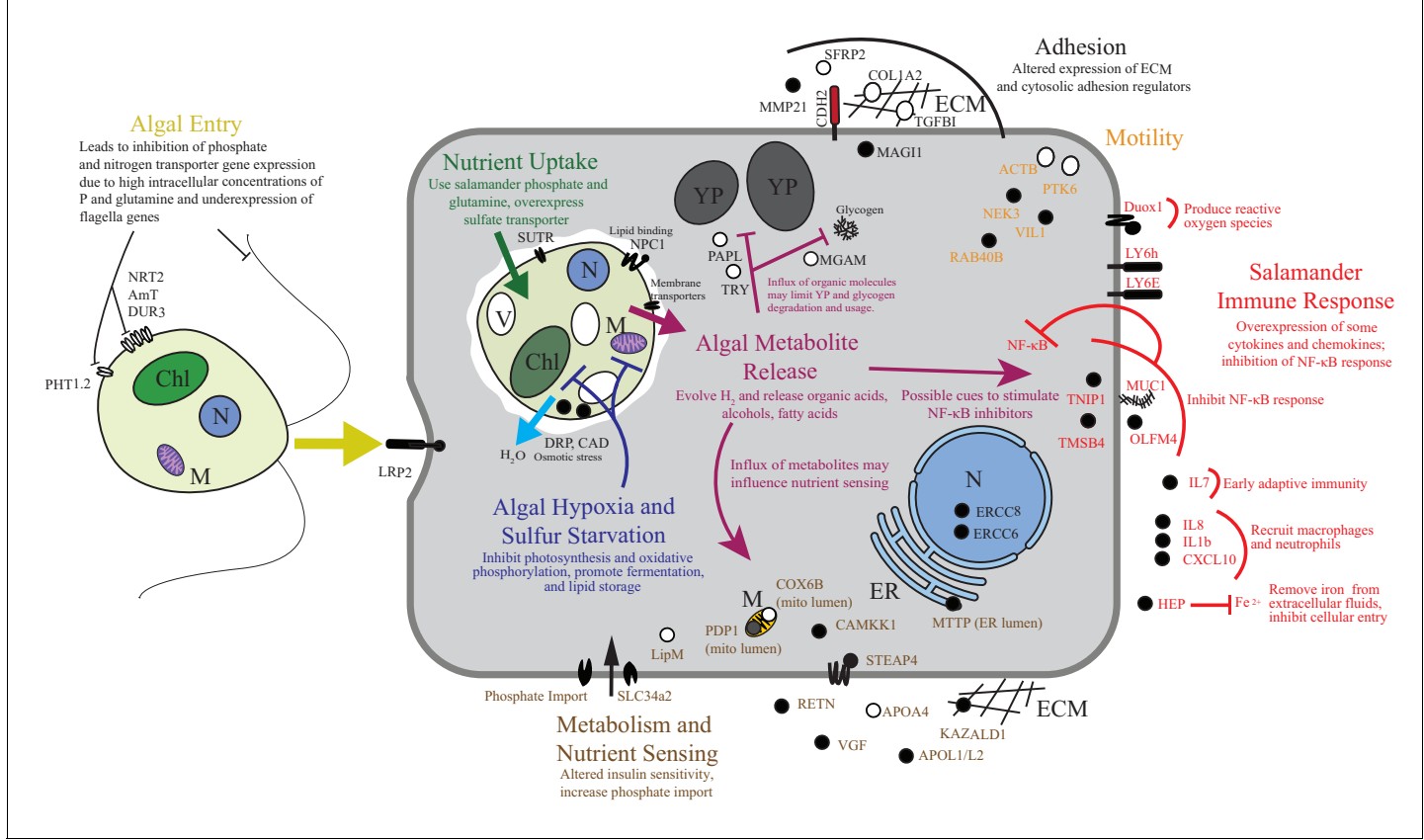

**Figure 6.** Summary of the major changes in both salamander and algal cells and how they may relate to one another. The inferred salamander responses are broken into four functional categories while algal changes fall within three primary functional categories based on gene annotations. Text indicates hypothetical changes within each category based on the implied roles of under-expressed or over-expressed genes. Major sections are color-coded. Over-expressed genes represented by solid black symbols. Under-expressed gene symbols are white with black outlines. Cellular compartments are in italics. *M*=mitochondrion, *YP*=yolk platelet, *V*=vacuole, *N*=nucleus, *ECM*=extracellular matrix, *ER*=endoplasmic reticulum, *Chl*=chloroplast.

ability to interrogate this endosymbiotic dialogue on a molecular level by co-opting the dual RNA-seq approach established for parasitology research to a non-model mutualism.

## Materials and methods

### Embryo and alga collection, cell isolation, and cDNA library preparation

*A. maculatum* cells in 90% RNAlater were diluted to a 50:50 solution of cells in RNA later and amphibian-phosphate buffered saline (APBS; PBS + 25% $H_2O$ to match amphibian osmolarity of 225 ± 5 mOsm/L) because the 90% RNAlater solution was too viscous for single cell isolation. The solution was spread on a glass slide and inspected between the bright field and epi-fluorescence illumination with a Chlorophyll filter set. Fifty salamander cells with intracellular algae and fifty salamander cells without intracellular algae per individual embryo were separated by manual single cell isolation (mouth pipetting) from dissociated embryos with a hand-pulled borosilicate pipette connected to a rubber tube. Each cell was collected directly from 45% RNA later into a microcentrifuge tube containing 200 µL lysis buffer (Extraction Buffer (XB), PicoPure RNA extraction kit; ThermoFisher Scientific). Four biological replicates were collected from four different *A. maculatum* embryos from the same clutch. Three biological replicate samples of intracapsular algae from the intracapsular fluid of three eggs were also collected—by aspiration with a syringe and 23 gauge needle—for RNA extraction. Additional RNA was prepared from three unialgal strains of *Oophila* including UTEX LB3005 and LB3006, which were established previously (*Kim et al., 2014*). The third algal strain—isolated

from an egg clutch sampled in 2012 from Greenbrook Sanctuary (Palisades, NY)—unfortunately, was lost during a laboratory power-outage. Quadruplicate sampling was made for quantitative analyses of the LB3005 strain.

RNA was extracted from each sample using the PicoPure RNA extraction kit following the manufacturer's recommended protocol for 'RNA Extraction from Cell Pellets', starting with incubation of the lysate at 42°C for 30 min with the modification of adding an equal volume of 70% ethanol to the 200 μL of lysis buffer. RNA for the cultured algal strains was prepared by using a combination of TRIzol (Thermo Fisher Scientific) and a Qiagen RNeasy kit.

Whole cDNA libraries were prepared directly from total RNA using the SMARTer Ultra Low cDNA Kit–HV (Clontech, Mountain View, CA) according to the manufacturer's protocol for a starting sample of 50 cells. Sequencing libraries for the Illumina HiSeq 2500 platform were prepared from the whole cDNA libraries using the Nextera-XT kit (Illumina, San Diego, CA) according to the manufacturer's protocol with an input of 375 pg cDNA per sample and a final clean-up step based on an Ampure-XP (Beckman Coulter, Brea, CA) protocol with the modification of adding 0.75x volume of bead solution to cDNA sample. Sequencing took place at the New York Genome Center and on an Illumina HiSeq 2500 sequencer with 125 bp paired-end reads. Transcriptome libraries (Illumina's TruSeq RNA library) for the cultured algae were prepared and sequenced at Genome Quebec on HiSeq 2000 with 100 bp paired-end reads or at Cornell Weill Genomics Resources Core Facility on the MySeq platform with 75 bp paired end reads.

## De novo assembly

Eleven paired-end whole cDNA libraries with greater than ten million paired reads per library were processed for assembly. There were four libraries from *A. maculatum* cells without intracellular algae, and four paired libraries (from the same individuals) from *A. maculatum* cells with intracellular algae, and three libraries from motile intracapsular algae. Quality trimming of the reads was performed with Trimmomatic (v 0.32) (*Bolger et al., 2014*) to remove low quality bases and adapter sequences. All eleven paired end libraries were used to construct a total evidence assembly using the Trinity algorithm (version trinityrnaseq_r20140717) (*Grabherr et al., 2011*; *Haas et al., 2013*). Transcriptomes of cultured algal strains were assembled separately in Trinity.

## Total evidence assembly filtering

The total evidence assembly returned 1,533,193 unique RNA-seq contigs that were clustered into 1,345,464 potential gene level (isoforms collapsed) transcript groups. The assembly largely consisted of a mixture of *A. maculatum* and *O. amblystomatis* transcripts. There were also 7,193 transcript groups (0.5%) corresponding to a predatory mite, *Metaseiulus occidentalis*, and 2,641 transcript groups corresponding to a dermal fungus, *Malassezia globosa*. Sequences corresponding to the mite and fungus were removed by BLASTN homology search (all BLAST analysis was completed using BLAST+ algorithms, v 2.2.28+ (*Camacho et al., 2009*) using BLAST databases comprised of all known transcript sequences from the genera Metaseiulus and Malassezia. Transcripts corresponding to the alga *O. amblystomatis* and salamander *A. maculatum* were recovered by BLASTN against a database consisting of transcript sequences from lab grown cultures of *O. amblystomatis* and sequences from the model salamander *Ambystoma mexicanum* (contributed by Randall Voss—University of Kentucky, and from [*Stewart et al., 2013*; *Wu et al., 2013*]). Transcripts were further filtered by a BLASTX homology search against a database containing the entire protein complement of: *Arabidopsis thaliana*, *Chlamydomonas reinhardtii*, *Mesostigma viride*, *Micromonas pusilla*, *Ostreococcus tauri*, *Oryza sativa*, *O. amblystomatis*, *Chrysemys picta bellii*, *Xenopus tropicalis*, *A. mexicanum*, *Pseudozyma*, *Saccharomyces cerevisiae*, and the genera *Melanopsichium and Leptosphaeria*. The assortment of species was chosen due to phylogentic proximity to *O. amblystomatis* or *A. maculatum*, or due to best hits from those genera/species found when a selection of transcripts was queried against the nr database. Best hits to plant or green algal species were noted as algal sequences and combined with the results of BLASTN against the *O. amblystomatis* database. Best hits to salamander or other animal species were noted as salamander sequences and combined with the results of BLASTN against the A. *mexicanum database*. Best hits to fungal sequences were discarded. The remainder with no known homology were retained and included as putative algal or salamander transcripts based on their expression pattern across samples.

One expectation of differential expression analysis is that most genes are expressed equally between control and experimental samples. If expression levels are ordered from low to high expression in control samples and binned in a sliding window of 100 genes per bin, the median expression level in each bin will increase as the index increases. Based on the expectation of equal expression, for the same sets of genes, the median expression level of experimental samples should correspondingly increase.

For a subset of genes confirmed to belong to *A. maculatum* by BLAST homology, genes were sorted by Fragments Per Kilobase of transcript per Million mapped reads (FPKM) values in the samples of *A. maculatum* without intracellular algae. Median FPKM values of bins of 100 genes, in a sliding window from the lowest expressed gene through the 100 genes with the highest FPKM values were calculated for the samples of *A. maculatum* with and without intracellular algae. The median FPKM values for the bins of *A. maculatum* only samples were plotted against the data from the samples of *A. maculatum* with intracellular algae (*Figure 1—figure supplement 2a*). At moderate to high expression levels, *A. maculatum* with intracellular alga FPKM values increased with *A. maculatum* without intracellular algae FPMK values (*Figure 1—figure supplement 2a*). But at very low FPKM values, the data were essentially uncorrelated. Median *A. maculatum* without intracellular alga FPKM values increased (since the data was ordered by those values), but median *A. maculatum* with intracellular alga FPKM values stayed the same (*Figure 1—figure supplement 2a*).

Lower limit FPKM values were determined by finding the FPKM value above which *A. maculatum* without intracellular alga samples and *A. maculatum* with intracellular alga samples exhibited a positive correlation (*Figure 1—figure supplement 2a*). Genes with FPKM values below the lower limit of both sets of samples being analyzed for differential expression (i.e. intracapsular algae and intracellular algae) were not included in the analysis. The uncorrelated expression pattern of genes with FPKM values below the threshold suggests that there is either insufficient sequencing depth to compare those genes between the two conditions, or those lowly expressed genes are expressed stochastically in these cells and the fluctuations in expression levels of those genes are not indicative of a biological difference between conditions. The same analysis was completed for *A. maculatum* cells with intracellular algae by ordering the genes based on their expression levels (*Figure 1—figure supplement 2b*), and for intracapsular (*Figure 1—figure supplement 2c*) and intracellular algal samples (*Figure 1—figure supplement 2d*).

The lower limit FPKM values for *A. maculatum* genes were 0.55 FPKM for *A. maculatum* cells (*Figure 1—figure supplement 2a*, vertical red-dashed line) and 0.61 FPKM for *A. maculatum* cells with algal endosymbionts (*Figure 1—figure supplement 2b*, vertical red-dashed line). The lower limit FPKM values for algal genes were 2 FPKM for the intracapsular alga (*Figure 1—figure supplement 2c*, vertical red-dashed line) and 0.04 FPKM for the intracellular alga (*Figure 1—figure supplement 2d*, vertical red-dashed line). The values are reflective of the sequencing depth of each sample, and are close to the widely used FPKM >1 lower limit threshold used in many RNAseq studies (*Fagerberg et al., 2014*; *Shin et al., 2014*; *Graveley et al., 2011*), except for the intracellular alga samples which suffer from low sequencing depth, but nonetheless display correlated expression with intracapsular alga samples starting at low FPKM values.

After determining the lower limit thresholds, the algal gene set consisted of genes with at least one read pair mapping to each of the three intracapsular algal samples *or* each of the four endosymbiotic cell samples that additionally were not found in the salamander, fungal, or mite BLAST data. Additionally, the genes had to have expression values above the lower limits described above in respective algal samples and below the lower limit for *A. maculatum* without intracellular alga samples in the salamander only cell samples for those genes. This resulted in a set of 8989 potential algal genes. However, due to a low depth of sequencing of the algal component of the endosymbiotic cell samples, additional filtering was necessary.

Genes that were not detected in intracellular algae could have been missing due to the lower depth of sequencing rather than representing an actual biological difference in expression between the algal populations. To determine what level of expression in intracapsular alga would be needed for a complete absence of measured expression in intracellular alga to be meaningful, the 8,989 algal genes were first ordered by intracapsular algal FPKM. Then the proportion of genes with no expression in intracellular samples was plotted against the median expression level in high-sequencing-depth intracapsular algae in bins of 100 genes in the ordered data. At low FPKM values in the intracapsular algae, up to 58% of the genes were absent from intracellular samples (*Figure 1—*

*figure supplement 3b*). As expression in intracapsular alga samples increased, the proportion of genes with measurable expression levels in intracellular algal samples increased as well (*Figure 1— figure supplement 3b*). The same relationship was not observed for *A. maculatum* data sets, where the depth of sequencing between samples was approximately equal (*Figure 1—figure supplement 4*).

The dependence of gene detection in intracellular algal samples on FPKM level in intracapsular algal samples abated at intracapsular alga FPKM values where 95% or more genes could be detected in intracellular algal samples (*Figure 1—figure supplement 3b*, vertical red-dashed line). That expression level corresponded to 67.9 FPKM in intracapsular algal genes. After removing genes with no detectable expression in intracellular algae with expression levels below 67.9 FPKM in intracapsular algae, the dependence of gene detection on expression level in intracapsular algae was removed, and the anomalous peak of undetected genes was removed from the expression histogram (see: *Figure 1—figure supplement 3a and c*). This resulted in a set of 6,781 genes. Due to finding some genes with homology to anomalous organisms such as pine and beech trees without homologs in *C. reinhardtii* in the set of 6,781 genes (perhaps due to pollen in the low cell number samples), only genes with homologs in the lab strain *Oophila* transcriptomes were considered. The final set of algal genes used in differential expression analysis consisted of 6,726 genes.

## Differential expression analysis

For transcripts derived from wild-collected samples, read mapping and transcript count quantification was accomplished using Bowtie2 (*Langmead and Salzberg, 2012*) and RSEM (*Li and Dewey, 2011*) using default parameters. For transcripts derived from cultured alga, read mapping and transcript quantification was accomplished using BBmap (*Bushnell, 2016*) and Salmon (*Patro et al., 2015*) (respectively). The read mapping and quantification algorithms used for reads from the cultured alga were implemented due to divergence of the two algal strains. The transcripts common between the two strains were on average 95% similar. BBmap paired with Salmon allowed for relaxed mapping parameters that were able to map the reads to the transcriptome despite differences in sequence composition with increased sensitivity compared to Bowtie2 plus RSEM. Prior to differential expression analysis, data driven abundance and homology filtering was implemented to derive the final gene lists used in differential expression analysis. A detailed account of filtering procedures can be found in Supplementary Materials and methods under the heading 'total evidence assembly filtering'.

Differential expression analysis was conducted in R (version 3.1.2) (*R Core Team, 2013*) using the edgeR package (*Robinson et al., 2010*). Generalized linear models were used for data analysis on normalized count data. Initial normalization of data derived from wild collected samples (endosymbiont free and endosymbiont containing salamander cells, intracapsular algae, and intracellular algae), was performed by trimmed mean of M-values (TMM) library size scaling-normalization (*Robinson and Oshlack, 2010*). Incorporation of data from unialgal cultures into differential expression analysis required additional GC-content normalization of the libraries, due to differences in GC-bias introduced by the two different library preparation methods (SMARTer cDNA synthesis followed by Nextera-XT library preparation for the wild collected samples and TrueSeq library preparation for the RNA preparation from unialgal cultures) (*Figure 7*). Normalization of GC-content and transcript length biases were accomplished using conditional quantile normalization (CQN) (*Hansen et al., 2012*). CQN resolves GC content and transcript length biases by fitting a model that incorporates observed read counts and a covariate such as GC content, and calculates an offset that is used to remove the covariation of these confounding factors. Following cqn normalization, differential analysis between unialgal cultures and wild collected intracapsular or intracellular algae was completed in edgeR. Salamander-only and salamander-plus-alga samples from the same individual embryo were considered paired samples for statistical analysis. Differentially expressed genes were considered as those with an FDR adjusted p-value less than 0.05.

## Functional annotation

Functional annotation of *A. maculatum* and *O. amblystomatis* transcripts was accomplished by BLASTX of transcripts against the UniProt-SWISSProt curated database (*Gasteiger et al., 2001*). BLASTX results were filtered by 'homology-derived structure of proteins' (HSSP) score (*Rost, 2002*)

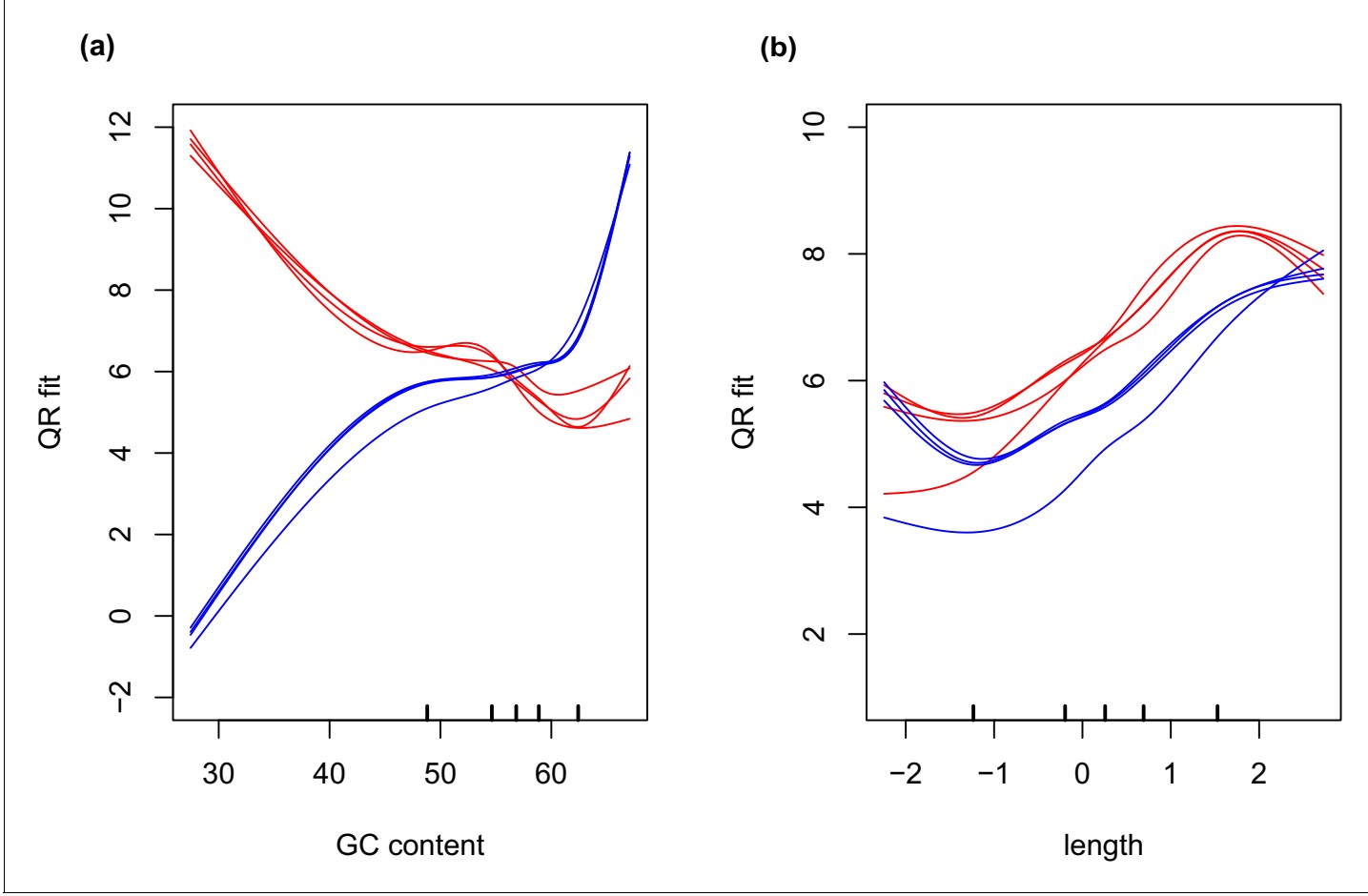

**Figure 7.** GC and transcript length bias in SMARTer-cDNA synthesis-Nextera-XT libraries compared to TrueSeq libraries. Red lines indicate the GC content or transcript length biases in reads obtained from SMARTer-cDNA synthesis-Nextera-XT libraries. Blue lines indicate the GC content or transcript length biases in reads obtained from TrueSeq libraries. (a) GC content and length are plotted against 'QRfit' which is a measure of fit by quantile regression to the models in *Hansen et al. (2012)*. This metric approximates bias in the sequence dataset by comparing read counts to expected models based on quantiles in the distribution of the GC content of the transcripts. The opposing trends in the two sets of lines shows that GC content bias between the two different libraries is vastly different. The reads obtained from SMARTer-cDNA synthesis-Nextera-XT libraries will tend to have more counts for low GC content transcripts, while the reads obtained from TrueSeq libraries will tend to have more counts for high GC content transcripts, systemically. (b) There is also some moderate transcript length bias differences between the two library prep methods visualized as the separation between the groups of red and blue lines. The methods implemented by the conditional quantile normalization (cqn) package in R handles both types of bias to make the gene count data from both library preparation methods comparable.

such that annotations were retained for hits with an HSSP_DIST score greater than 5 (*Burns et al., 2015*). UniProt ID annotations were assigned based on the maximum HSSP score for each gene. This led to 14,761 functional annotations for *A. maculatum* transcripts and 3,850 functional annotations of *O. amblystomatis* transcripts. Further annotation of differentially expressed transcripts was accomplished by HHblits (*Remmert et al., 2012*) homology detection. Unannotated, differentially expressed transcripts were translated in all six reading frames and the translations were processed by HHblits. Significant hits were determined by manual inspection of the HHblits output. Transposable elements were categorized by homology to known transposons by BLASTX or HHblits, and through the use of the PASTEClassifier tool (*Hoede et al., 2014*). Multiple alignments were created with MUSCLE using default parameters (*Edgar, 2004*). Alignments were visualized in SeaView (*Gouy et al., 2010*).

## Functional grouping of differentially expressed genes

Due to the novelty of this symbiosis, the non-model organisms under consideration, and the multi-organism functional annotation obtained, an initial automated functional grouping, gene ontology (GO) term analysis, completed as in *Burns et al. (2015)* (*Supplementary files 13* and *14* for algal and salamander GO term analysis, respectively) was determined to be insufficient to understand the likely biological function of over and under-expressed genes. Other tools such as REVIGO (*Supek et al., 2011*) performed better (*Supplementary files 15* and *16* for algal genes, and *Supplementary files 17* and *18* for salamander genes), but did not catch large functional modules that were evident upon further manual inspection of the gene lists. For the small sets of differentially expressed genes observed between salamander cells with or without algae, and between intracapsular and intracellular algae, manual curation of each differentially expressed gene was implemented by performing an extensive literature search for each of the differentially expressed genes, based on the SWISSProt or HHblits annotation. Relevant functions associated with each gene in the scientific literature were noted, and those functions were grouped manually to give the final functional categories discussed in the text. For the larger set of differentially expressed genes observed between algae cultured in nutrient replete media and intracellular algae, only automated annotation and GO term grouping (REVIGO) was used to define functional categories.

## qPCR validation of gene expression patterns in the alga *O. amblystomatis*

Hypotheses concerning gene expression patterns associated with endosymbiosis in the alga *O. amblystomatis* were tested using culture strains of the alga. Validation experiments could not be conducted in the salamander *A. maculatum* due to the seasonal nature of the association and the lack of any laboratory stock of *A. maculatum*.

*O. amblystomatis* cultures were maintained at 18°C under a 12 hr light/12 hr dark cycle with an average light intensity of 34 µmol·m$^{-2}$·sec$^{-1}$, in AF-6 medium (*Kato, 1982*) with modifications as described previously ( *et al., 2004*). For examining the dependence of phosphate transporter expression on phosphate levels in the media, AF6 media was formulated without potassium phosphate. Appropriate quantities of concentrated potassium phosphate at a ratio of 1:2 K$_2$HPO$_4$: KH$_2$PO$_4$ plus potassium chloride (to a final concentration of 130.8 µM K$^+$ ions, the concentration of K$^+$ ions present in normal AF-6 media) were added to the phosphate deficient AF-6 media to make AF-6 with the various phosphate levels used in the experiment. For examining the dependence of nitrogen-related transporters on nitrogen and glutamine concentrations, AF-6 media was formulated with various levels of nitrate and ammonia with the addition of NaCl to balance the loss of Na$^+$ ions from leaving NaNO$_3$ out of the media.

For phosphate limitation experiments, *O. amblystomatis* cells growing in AF-6 media were pelleted (1,000xg for 5 min) and washed three times with media completely depleted of phosphate. Following the third wash, cells were re-suspended in phosphate depleted media. Cells were counted and aliquoted into flasks at a concentration of 40,000 cells·mL$^{-1}$ in 5 mL total volume per flask of phosphate deplete AF-6. For each phosphate concentration, 5 mL of AF-6 media with 2x phosphate was added to the appropriate flask to give the appropriate phosphate concentration for the experiment. Cultures were grown for 5 days prior to harvesting for RNA purification. Three flasks of algae were assayed for each phosphate concentration (from 100 pM to 10 mM phosphate in 10-fold intervals).

For nitrogen limitation and glutamine experiments, *O. amblystomatis* cells were prepared as described for the phosphate limitation experiments using nitrogen depleted AF-6 media. Cultures were grown for 5 days at nitrogen concentrations approximating observed nitrate and ammonia concentrations in salamander egg capsules (6.6 µM NO$_3^-$; 17 µM NH$_4^+$) prior to addition of glutamine. Glutamine was added to a final concentration of 2 mM. Algal cells were harvested after 6 hr of incubation with or without glutamine and assayed for gene expression. Three flasks of algae were assayed for each condition.

To prepare cDNA, *O. amblystomatis* cells were harvested by centrifugation (1,000xg for 5 min), and 350 µL lysis buffer RLT-Plus with fresh $\beta$-mercaptoethanol (RNeasy mini Plus kit, Qiagen, Valencia, CA) was added, and the lysate was vortexed for 30s. RNA was purified from the lysate following the manufacturer's protocol. Purified RNA was converted to cDNA using the Quantitect RT kit

(Qiagen) following the manufacturer's protocol. Resultant cDNA was diluted with three volumes of 10mM Tris buffer, pH 7.5, or RNase free water prior to qPCR reactions.

Quantitative PCR primers for four reference genes and five response genes (*Table 3*) were designed using conserved regions in multiple sequence alignments of cDNA sequences from the three *O. amblystomatis* cultured strains as well as the *O. amblystomatis* sequences obtained from the field material. Candidate reference genes were selected due to their utility in prior studies in the related chlorophycean alga *Chlamydomonas reinhardtii,* and were confirmed to have equivalent transcriptome expression levels in intracapsular and intracellular alga in this study. The reference genes were RPL32 & H2B1 (*Liu et al., 2012*), RACK1 (*Mus et al., 2007*), and YPTC1 (*Lake and Willows, 2003*). Several primer pairs were designed for each reference and response gene using the software tools GEMI (*Sobhy and Colson, 2012*), Primer3 (*Untergasser et al., 2012*), and Primer-Quest (*Owczarzy et al., 2008*). Primer pairs were validated by making standard curves using a cDNA dilution series. Primer pairs with the lowest Cq value for a given gene and PCR efficiencies between 0.9 and 1.1 in a standard curve of cDNA dilution series were validated for use in gene expression studies (*Table 3*).

Quantitative PCR reactions used 1 μL of the diluted cDNA in 20 μL reactions with a 700 nM concentration of each primer using QuantiNova Sybr green (Qiagen) for amplification and detection. QPCR reactions were done in duplicate. Reactions were performed on a RotorGeneQ instrument (Qiagen) with a 2-step cycling program of 5s at 95°C and 10s at 60°C followed by melting curve analysis. Raw data was exported from the RotorGeneQ and per run-per amplicon efficiency correction was implemented in LinRegPCR (version 2015.3) (*Ramakers et al., 2003*; *Ruijter et al., 2009*). Differences in expression were analyzed using ANOVA with contrasts in R.

**Table 3.** *O. amblystomatis* qPCR primer sequences.
Primer pairs for four reference genes (*RACK1, YPTC1, RPL32, H2B1*), and five response genes (*PhT1.2, NaPhT1 [ANTR1], AMT1.2, NRT2.4, DUR3*) used in this study. Efficiency values were measured per amplicon using a standard curve with five two-fold dilutions of cDNA.

| Primer | Sequence (5'−3') | Efficiency |
|---|---|---|
| Ooph_RACK1_L_3 | CGCACAGCCAGTAGCGGT | 0.94 |
| Ooph_RACK1_R_3 | GGACCTGGCTGAGGGCAA | |
| Ooph_YPTC1_L_4 | TTGCGGATGACACCTACACG | 1.09 |
| Ooph_YPTC1_R_4 | TGGTCCTGAATCGTTCCTGC | |
| Ooph_RPL32_L_2 | ATAACAGGGTCCGCAGAAAG | 1.03 |
| Ooph_RPL32_R_2 | GTTGGAGACGAGGAACTTGAG | |
| Ooph_H2B1_L_4 | CAAGAAGCCCACCATGACCT | 1.04 |
| Ooph_H2B1_R_4 | GGTGAACTTGGTGACTGCCT | |
| Ooph_PhT1.2_L_4 | TGCCAATGACTTCGCCTTCT | 1.02 |
| Ooph_PhT1.2_R_4 | ACGTTCCACTGCTGCTTCTT | |
| Ooph_NaPhT1_L_4 | TCCATCATCGGTCTGTCGCT | 0.99 |
| Ooph_NaPhT1_R_4 | GAACCACACGATGCCCAGAG | |
| Ooph_AMT1.2_L_4 | CGGTCTCCTTCCAATCGCCA | 0.96 |
| Ooph_AMT1.2_R_4 | CCAATGGGTGCTGACTGGGA | |
| Ooph_NRT2.4_L_3 | CGACTACCGCGACCTGAAGA | 1.03 |
| Ooph_NRT2.4_R_3 | GAACAAGACCCAGGCCCTGT | |
| Ooph_DUR3_L_3 | GCGAATGCCGAGCACTTC | 1.02 |
| Ooph_DUR3_R_3 | CTGTCCCTGGGCTGGGT | |

## Ethics approval and consent to participate

The Institutional Animal Care and Use Committee of Gettysburg College approved the research on salamander embryos (IACUC#2013 F17). Field collection of egg masses was completed under Pennsylvania Fish and Boat Commission permit (PA-727 Type 1).

## Availability of data and material

All transcriptome assemblies and read data are available from the NCBI transcriptome shotgun assembly database under BioProject #PRJNA326420. Other relevant data are within the paper and its additional files.

# Acknowledgements

We thank S Obado, M Bradic, S Thurston for technical comments and critical reviews and R Voss for providing the unpublished V4 assembly of an axolotl (*Ambystoma mexicanum*) transcriptome. This work was supported in part by a grant to Gettysburg College from the Howard Hughes Medical Institute through the Precollege and Undergraduate Science Education Program. The work was supported by an NSF EAGER to R. Kerney, E Kim, and JA Burns (#1428065) and NSF CAREER to E Kim (#1453639). The funders had no role in study design, data collection and analysis, decision to publish, or preparation of the manuscript.

# Additional information

### Funding

| Funder | Grant reference number | Author |
|---|---|---|
| National Science Foundation | 1428065 | John A Burns<br>Eunsoo Kim<br>Ryan Kerney |
| National Science Foundation | 1453639 | Eunsoo Kim |

The funders had no role in study design, data collection and interpretation, or the decision to submit the work for publication.

### Author contributions

JAB, Conceptualization, Resources, Data curation, Software, Formal analysis, Supervision, Funding acquisition, Validation, Investigation, Visualization, Methodology, Writing—original draft, Project administration, Writing—review and editing; HZ, EH, Formal analysis, Investigation; EK, Conceptualization, Resources, Formal analysis, Supervision, Funding acquisition, Investigation, Methodology, Writing—original draft, Project administration, Writing—review and editing; RK, Conceptualization, Formal analysis, Supervision, Funding acquisition, Validation, Investigation, Methodology, Writing—original draft, Project administration, Writing—review and editing

### Author ORCIDs

John A Burns, http://orcid.org/0000-0002-2348-8438

### Ethics

Animal experimentation: The Institutional Animal Care and Use Committee of Gettysburg College approved the research on salamander embryos (IACUC#2013F17). Field collection of egg masses was completed under Pennsylvania Fish and Boat Commission permit (PA-727 Type 1).

# Additional files

### Supplementary files

• Supplementary file 1. Differentially expressed stress related genes in *O. amblystomatis*

• Supplementary file 2. Differentially expressed sulfur metabolism genes in *O. amblystomatis*

• Supplementary file 3. Differentially expressed genes with roles in fermentation in *O. amblystomatis*

• Supplementary file 4. Differentially expressed genes in photosynthesis in *O. amblystomatis*

• Supplementary file 5. Differentially expressed genes in mitochondrial electron transport in *O. amblystomatis*

• Supplementary file 6. Differentially expressed nitrogen and phosphorous transport genes in *O. amblystomatis*

• Supplementary file 7. Differentially expressed transposable element genes in *A. maculatum*

• Supplementary file 8. Differentially expressed proliferation genes in *A. maculatum*

• Supplementary file 9. Differentially expressed genes with immune functions in *A. maculatum*.

• Supplementary file 10. NF-κB and TLR response gene expression levels in salamander cells with algal endosymbionts.

• Supplementary file 11. Differentially expressed genes in metabolism and nutrient sensing in *A. maculatum*.

• Supplementary file 12. Differentially expressed genes in motility in *A. maculatum*

• Supplementary file 13. Top 25 biological process GO annotations for differentially Expressed *O. amblystomatis* genes.

• Supplementary file 14. Top 25 biological process GO annotations for differentially expressed *A. maculatum* genes.

• Supplementary file 15. Top 25 biological process GO annotations from REViGO for differentially expressed *O. amblystomatis* genes.

• Supplementary file 16. Functional grouping of *O. amblystomatis* genes by REViGO.

• Supplementary file 17. Top 25 biological process GO annotations from REViGO for differentially expressed *A. maculatum* genes.

• Supplementary file 18. Functional grouping of *A. maculatum* genes by REViGO.

## Major datasets

The following dataset was generated:

| Author(s) | Year | Dataset title | Dataset URL | Database, license, and accessibility information |
|---|---|---|---|---|
| Burns JA, Zhang H, Hill E, Kim E, Kerney RK | 2016 | Mechanisms of Establishing and Maintaining an Algal Endosymbiont in a Vertebrate Host | https://www.ncbi.nlm.nih.gov/bioproject/PRJNA326420 | Publicly available at the NCBI BioProject database (accession no: PRJNA326420) |

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
