## [Decision Letter]

Thank you for submitting your article "Transcriptome Analysis Illuminates the Nature of the Intracellular Interaction in a Vertebrate-Algal Symbiosis" for consideration by *eLife*. Your article has been reviewed by two peer reviewers, and the evaluation has been overseen by a Reviewing Editor and Ian Baldwin as the Senior Editor. The following individual involved in review of your submission has agreed to reveal his identity: Senjie Lin (Reviewer #2).

The reviewers have discussed the reviews with one another and the Reviewing Editor has drafted this decision to help you prepare a revised submission.

Summary:

This paper reports a very interesting study of the *Oophila amblystomatis* (green alga) – *Ambystoma maculatum* (salamander) symbiosis. Using RNA-seq the authors compared the green alga egg capsule ectosymbiont transcript repertoire to those of the endosymbionts as well as when the alga was grown in replete culture conditions. In addition, they compared the transcriptome of the salamander cells when infected or not by the green algal symbiont. The importance of this work is that it provides a unique perspective into the physiological consequences of the intracellular environment that are faced by algal endosymbionts.

The low level of differential expression for the ecto- versus the endosymbiont state of the alga (ca. 4%) and that of infected versus uninfected salamander cells (ca. 0.64%) is surprising, but not perhaps as surprising as when the algal data are compared to these cells in culture, which differ over 40% of the observed transcripts. This leads the reviewers to conclude that the *Oophila* – *Ambystoma* ecto- and endosymbiosis model provides valuable insights into the changes that typify endosymbiosis of algae in animals, as well as provides a useful window into endosymbiosis in general. In addition to these considerations, the work results in novel genome data for future studies.

Essential revisions:

The authors have convincingly demonstrated that the algae are facing anoxic conditions that cannot be compensated by photosynthetic production of oxygen. In addition to this, the authors provide convincing data that, by virtue of modification in nitrogen and phosphate transport that the "symbiont" is tapping into the glutamine and phosphate cytosolic stores without compensating by supplying oxygen through photosynthesis. Yet oxygen supply to the oxygen hungry salamander cells has been demonstrated to be a major determinant of the *Oophila* – *Ambystoma* ectosymbiosis in the egg capsule. In addition, there is no evidence that the endosymbionts are exporting significant amounts of photosynthate, in agreement with the general "fermentative metabolism" status of the alga. Reduced photosynthesis is also backed by reduced PSII and by the fact that light might become limiting within the mass of embryo cells. In *Chlamydomonas* plastid starch is the major substrate fermented under such anoxic conditions in darkness. It is suggested that staining the embryo cells with iodine solution would allow the authors to easily monitor through microscopy the presence or absence of the polysaccharide. If starch disappears it would strengthen their hypothesis that photosynthesis is either not operating in these algae or operating at a very low level and that carbohydrates or lipids are unlikely to be supplied to the host in significant amounts.

One reviewer disagreed that secretion of fermentation end products are beneficial to the host, in particular if such products are produced under "heterotrophic" conditions in an anoxic environment. In large amounts, these products, are potentially either cytotoxic by themselves (ethanol) or will impact negatively the osmotic pressure of the cytosol. This leaves us no clues as to what are the beneficial aspects of this endosymbiosis. There is no evidence supporting vertical transmission of the symbiont so maintenance of the alga for "infection" of next generation is not relevant. It is suggested that the authors revise their

Discussion to further emphasize these inconsistencies when they describe areas for future research. Another possibility is that the endosymbionts are supplying additional metabolites in small amounts, in addition to oxygen by the ectosymbionts. Because ectosymbionts are used as controls here to fish for the endosymbiosis specific response of the algae, major modifications in those pathways concerned with such metabolites would escape detection. The authors need to discuss this issue as well as consider vitamin or amino acid supply. In this regard, the reviewers wanted an explicit explanation for the simultaneous metabolic shift toward fermentation and oxidative stress conditions, because the former suggests insufficient O2 and the latter, excess. It seems that the salamander's potential to produce ROS described in the paper would fit the scenario but it was not explicitly discussed in the current version.

---

## [Author Response]

*Essential revisions:*

*The authors have convincingly demonstrated that the algae are facing anoxic conditions that cannot be compensated by photosynthetic production of oxygen. In addition to this, the authors provide convincing data that, by virtue of modification in nitrogen and phosphate transport that the "symbiont" is tapping into the glutamine and phosphate cytosolic stores without compensating by supplying oxygen through photosynthesis. Yet oxygen supply to the oxygen hungry salamander cells has been demonstrated to be a major determinant of the Oophila – Ambystoma ectosymbiosis in the egg capsule. In addition, there is no evidence that the endosymbionts are exporting significant amounts of photosynthate, in agreement with the general "fermentative metabolism" status of the alga. Reduced photosynthesis is also backed by reduced PSII and by the fact that light might become limiting within the mass of embryo cells. In Chlamydomonas plastid starch is the major substrate fermented under such anoxic conditions in darkness. It is suggested that staining the embryo cells with iodine solution would allow the authors to easily monitor through microscopy the presence or absence of the polysaccharide. If starch disappears it would strengthen their hypothesis that photosynthesis is either not operating in these algae or operating at a very low level and that carbohydrates or lipids are unlikely to be supplied to the host in significant amounts.*

The reviewers’ comment on the use of starch in anoxic fermentation is a terrific insight and one that we have coincidentally analyzed using TEM in our 2011 paper. In our earlier analysis we found that the starch granules do not differ in size when compared between the intracapsular and intracellular algal cells, however starch does make up more of the intracapsular algal cytoplasm:

"Several changes to algal cells were associated with host tissue and cellular invasion. Relative area measurements of starch granules and vacuoles were compared between intracapsular (n = 26), intracellular (n = 24), and extracellular algae found within the embryonic alimentary canal (n = 18). There were no differences in starch granule sizes [one-way ANOVA, F(2,56) = 2.89, P > 0.05]; however, starch granules constituted more cross-sectional area of intracapsular algae (10.5% SD 9.2%) than intracellular (5.9% SD 4.2%) or alimentary canal (3.4% SD 3.2%) algae [one-way ANOVA, F(2,56) = 4.70, P < 0.05]."(Kerney et al. 2011)

Please see the inserted paragraph on page 12, lines 243-255 where we describe these starch granule results and the consistent role of sulfur deprivation in initiating fermentation in other Chlamydomonad algae:

“The occurrence of intracellular fermentation is also supported by decreased starch granules in the intracellular algae and transcriptional changes to algal sulfate transport and sulfur metabolism associated genes. […] The overexpression of a sulfate transporter and taurine catabolic enzymes (Chader, Hacene and Agathos, 2009), along with other transcripts associated with sulfur metabolism (Figure 3; Table 4), indicates that fermentation in intracellular *Oophila* coincides with sulfur deprivation and closely matches the consumption of starch found in other fermenting algae.”

*One reviewer disagreed that secretion of fermentation end products are beneficial to the host, in particular if such products are produced under "heterotrophic" conditions in an anoxic environment. In large amounts, these products, are potentially either cytotoxic by themselves (ethanol) or will impact negatively the osmotic pressure of the cytosol. This leaves us no clues as to what are the beneficial aspects of this endosymbiosis. There is no evidence supporting vertical transmission of the symbiont so maintenance of the alga for "infection" of next generation is not relevant. It is suggested that the authors revise their Discussion to further emphasize these inconsistencies when they describe areas for future research.*

We have inserted a new section to the Discussion titled: “The Nature of the Endosymbiosis”. We believe this additional text addresses these concerns raised by our reviewers:

“The Nature of the Endosymbiosis

Whether the alga benefits from this endosymbiotic interaction remains unclear. Similar questions of net “mutualism” persist for the symbiosis between the bobtail squid *Eprymna scolopes* and the bacterium *Allivibrio fischeri* (McFall-Ngai, 2014) although in both systems microbial cells exhibit specific taxic responses to a developing host, suggesting an adaptive origin of these behaviors. […] Whether the intracellular algae are on the positive end of a net host benefit remains uncertain, however it is clear that the algae have an unconventional “photosymbiont” role."

*Another possibility is that the endosymbionts are supplying additional metabolites in small amounts, in addition to oxygen by the ectosymbionts. Because ectosymbionts are used as controls here to fish for the endosymbiosis specific response of the algae, major modifications in those pathways concerned with such metabolites would escape detection. The authors need to discuss this issue as well as consider vitamin or amino acid supply. In this regard, the reviewers wanted an explicit explanation for the simultaneous metabolic shift toward fermentation and oxidative stress conditions, because the former suggests insufficient O2 and the latter, excess. It seems that the salamander's potential to produce ROS described in the paper would fit the scenario but it was not explicitly discussed in the current version.*

To address the question about ROS stress and fermentation, we added text about intracellular parasites (primarily *Plasmodium*) and the ectosymbiont mutualist *Aliivibrio fischeri* fermenting in the intracellular and light organ environments, respectively, in the section "The Nature of the Endosymbiosis"(see above), and on how ROS can be generated under hypoxia:

“Moreover, we observed under-expression of subunit 6b of cytochrome c oxidase (COX6B1). Reduction of cytochrome c oxidase activity is associated with the generation of ROS through signaling to endoplasmic reticulum NADPH oxidases in yeast (Leadsham et al., 2013). […] Importantly, the generation of ROS does not rely on the presence of high levels of oxygen in vertebrate cells as ROS can be generated in hypoxic as well as normoxic conditions through a variety of mechanisms (Kim et al., 2011; Nathan and Cunningham-Bussel, 2013) although there is no indication that the salamander cells are suffering from hypoxia here.”

We don't intend to suggest the salamander cells are hypoxic, as we don't have evidence for that, and have clarified in the text above. However, we do want to point out that hypoxia does not preclude generation of ROS as the reviewer has suggested above.